# Zero in on Faithful Anchors: High-Fidelity Visual Token Condensation for Multimodal Large Language Models

## Abstract

Multimodal Large Language Models (MLLMs) have demonstrated impressive visual reasoning capabilities, but their scalability is limited by the computational burden of processing massive visual tokens. To alleviate this bottleneck, many studies have focused on visual token pruning strategies, which utilize cross-attention or [CLS] attention to identify and retain informative visual tokens. In this work, we uncover a critical limitation of such pruning approaches, *i.e.*, they tend to either omit or pay much attention to the background context within images, resulting in potential semantic distortion. To solve this problem, we introduce CondenseVLM, a dynamic token compression framework for HiFi and efficient MLLM inference, that enhances the information density of retained visual tokens. In particular, CondenseVLM employs a three-stage method: it first selects high-attention tokens as faithful anchors to preserve fine-grained semantics, then compensates important background tokens, and finally merges the retained tokens based on spatial proximity and semantic similarity to ensure view integrity. This synergistic optimization of semantic uniqueness, spatial coverage, and contextual integrity makes CondenseVLM capable of high-fidelity compression. Extensive experiments demonstrate that CondenseVLM can prune up to 77.8% with just a 1.2% drop. Moreover, it integrates seamlessly with efficient attention operators during decoding, delivering substantial speedups and memory savings. *The code will be released.*

## 1 Introduction

Recent advances in Multimodal Large Language Models (MLLMs) have achieved remarkable success in various vision-language tasks (Liu et al., 2024c; Wang et al., 2025; 2024b). However, their practical deployment faces significant challenges due to substantial computational demands (Liu et al., 2024a). A primary contributor to this computational overhead stems from the excessive quantity of visual tokens processed during inference, particularly when handling high-resolution images or multi-frame video inputs (Lin et al., 2023). For example, LLaVA-1.5 (Liu et al., 2024a) processes 576 visual tokens per image, a volume that typically exceeds the number of textual tokens by an order of magnitude. Empirical evidence suggests that conventional grid-based visual tokenization strategies often capture *substantial redundant* information from *low-information-density background regions* (Achiam et al., 2023; Chen et al., 2023; Dosovitskiy et al., 2021). This observation underscores the need for adaptive token condensation to eliminate redundancy and preserve task-relevant information.

To address this issue, numerous token pruning or merging approaches have been proposed to reduce visual token quantities for efficient MLLM inference in a training-free manner. For instance, FastV analyzes the information flow in MLLMs (Chen et al., 2024), revealing that visual tokens receive significantly less attention than text tokens in deeper LLM layers. Consequently, they propose ranking visual tokens by their text-visual attention scores and pruning those with lower scores to mitigate redundancy in LLMs and enhance the model's inference efficiency. More works (Shang et al., 2024; Ye et al., 2025; Zhang et al., 2024b) employ similar strategies to prune redundant visual tokens through text-visual attention, but they fail to address the dispersion of attention (Zhang et al., 2024a).

In contrast to the aforementioned approaches, alternative strategies such as FasterVLM (Zhang et al., 2024a) and Vision-Zip (Yang et al., 2024a) adopt vision-centric paradigms to identify salient visual

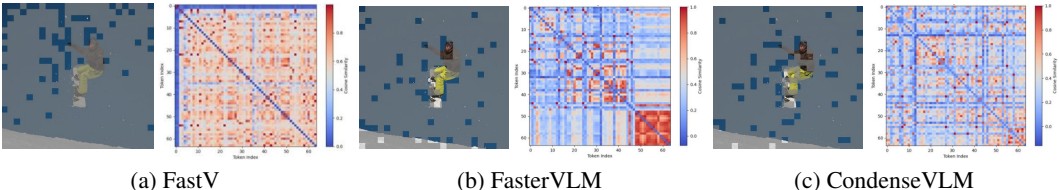

Figure 1: Comparison of text-centric pruning paradigm FastV, and vision-centric scheme FasterVLM.

tokens. As visualized in Fig. 1, FasterVLM leverages the attention weights associated with the [CLS] token in the visual encoder to quantify the importance of individual visual tokens. By pruning these tokens before feeding them into the large language model (LLM), these methods seek to reduce redundancy in the image token sequence. This pruning strategy yields enhanced performance and accelerated inference compared to FastV (Chen et al., 2024): notably, [CLS] attention encodes *holistic representations* of visual content, enabling more precise identification of low-information visual tokens. In this work, we critically analyze the key limitations inherent to these pruning approaches.

|  |  |  |
|---|---|---|
| (a) FastV | (b) FasterVLM | (c) CondenseVLM |

Figure 2: Comparison of similarity map between the retained visual tokens in different methods.

▷ **Redundancy in high-attention tokens**. Relying solely on attention scores for token selection often results in the preservation of clusters of similar tokens, leading to information redundancy and inefficient utilization of computational resources. To illustrate this issue, we pruned 88.9% of the tokens using FastV, FasterVLM, and then computed the cosine similarity matrix between the remaining tokens. The visualizations in Fig. 2 highlight a significant difference between the methods. Approaches that rely exclusively on attention (FastV and FasterVLM) exhibit a higher degree of similarity between the remaining tokens. This overlap in features makes it challenging for attention to discern unique or critical information and also impedes the diversity of visual representations.

▷ **Spatial distribution of attention**. [CLS] attention often exhibits a highly concentrated distribution, as illustrated in Fig. 3a, with a small subset of visual tokens receiving the majority of attention, and these high-attention tokens usually serve with informativeness, as the [CLS] token is trained to aggregate global information of an image. Thus, these attention-first methods create a spatial bias, causing them to systematically ignore tokens from other low-attention regions that provide crucial contextual information essential for a comprehensive sense understanding.

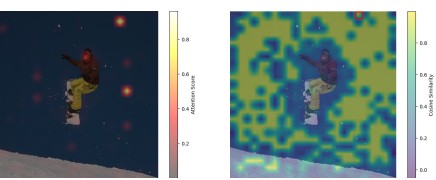

|  |  |
|---|---|
| (a) [CLS] attention | (b) Similarity map |

Figure 3: (a) Distribution of [CLS] attention map. (b) Similarity of different tokens with the background token (marked in red box).

▷ **Semantic disruption in token merging**. Existing token merging methods rely exclusively on feature similarity, which is insufficient due to inherent limitations in visual representation spaces. As evidenced in Fig. 3b, the feature embedding space often assigns high similarity scores to tokens from spatially distant regions that share visual characteristics (e.g., extensive homogeneous areas typically perceived as 'sky' or 'snow'), which creates a fundamental misalignment: *feature proximity fails to consistently reflect semantic relationships or spatial coherence within the image*. Consequently, similarity-driven merging frequently produces suboptimal groupings, *i.e.*, fusing semantically distinct regions while separating semantically related ones—ultimately corrupting the integrity of the visual representation. These findings motivate us to resolve this misalignment through a spatial-semantic merging strategy that jointly optimizes both visual feature similarity and spatial coherence.

To address these limitations, we propose CondenseVLM, a dynamic token compression framework that synergistically optimizes feature uniqueness, spatial coverage, and information integrity. Our approach comprises two key stages: First, we select a predefined number of anchor tokens, consisting of high-information-density tokens (*characterized by high attention scores and low cosine similarity*

*to each other*) and high-global-similarity background tokens. After anchor selection, we cluster the remaining tokens into groups based on both cosine similarity and spatial distance to the anchors, followed by an attention-weighted fusion process. This streamlined process maintains computational efficiency while surpassing prior methods in performance metrics. The main contributions include:

❶ *Comprehensive analysis of existing strategies*. We present a thorough analysis, supported by experimental evidence, of the limitations inherent in both attention-based token selection and similarity-based token merging strategies. This analysis highlights the critical need for a more nuanced approach that addresses the shortcomings of these conventional methods.

❷ *An effective token compression framework*. We introduce CondenseVLM, a plug-and-play visual token merging strategy for efficient MLLM inference without additional fine-tuning. CondenseVLM distinguishes itself by strategically incorporating both feature diversity and spatial relationships, enabling a more informed and effective visual token condensation.

❸ *Demonstrated Efficiency-Performance Trade-off*. Extensive experimental results demonstrate that CondenseVLM achieves a compelling balance between computational efficiency and performance. For instance, when applied to the LLaVA-1.5-7B model, CondenseVLM achieves an impressive token reduction rate of 77.8%, while incurring a performance decrease of only 1.2%. This showcases the practical effectiveness of CondenseVLM in real resource-constrained scenarios.

## 2 MOTIVATION

Building on the limitations of previous attention-based token pruning methods outlined in Sec. 1, we identify two key shortcomings. Regarding *token selection*, those relying on attention methods exhibit a high degree of similarity (as shown in Tab. 1) and neglect critical background context among selected tokens, which impedes feature diversity, limiting comprehensive scene understanding. In terms of *token merging*, indiscriminate merging of residual tokens based solely on feature similarity can lead to suboptimal token merging, compromising the accuracy of the visual representation.

Table 1: The proportion of similar tokens (cosine similarity $\geq \gamma$) among the selected visual tokens.

| FastV | FasterVLM | CondenseVLM |
|---|---|---|
| 9.92% | 10.71% | 4.32% $\Delta$ -5.6% |

These limitations motivate us to explore principled strategies for effectively selecting multifaceted, salient tokens while systematically merging residual tokens. Inspired by the superpixel segmentation algorithm (Achanta et al., 2012), we propose three mechanisms to address the previous shortcomings:

SIMILARITY SUPPRESSION. We maintain feature diversity by suppressing inter-token similarity within selected tokens, which ensures the retained tokens capture a broader spectrum of images.
CONTEXTUAL TOKEN SELECTION. To mitigate excessive loss of visual context, we explicitly incorporate background tokens into selection, which enables balanced representations to be learned.
SPATIAL-SEMANTIC AWARE MERGING. By integrating both semantic similarity and spatial distance into the token merging framework, we ensure that fused tokens exhibit both semantic relevance and positional coherence. The semantic integrity of the visual presentation is preserved in this way.

## 3 METHODOLOGY

In this section, we present CondenseVLM, a dynamic token compression framework designed to accelerate multimodal reasoning by strategically merging redundant visual tokens. As illustrated in Fig. 4, CondenseVLM operates prior to the LLM decoder and consists of three core stages: *Faithful Anchor Selection*, *Background Token Selection*, *Spatial-Similarity Contextual Merging*. The token selection process retains $R\%$ of the total tokens, where $\alpha R\%$ are designated as visual anchor tokens and $(1 - \alpha)R\%$ are designated as background tokens, and $\alpha \in [0, 1]$ governs the allocation.

### 3.1 FAITHFUL ANCHOR SELECTION

To identify key visual tokens as anchors, we evaluate the importance of each visual token based on [CLS] attention scores obtained from the visual encoder (Zhang et al., 2024a). Specifically, the attention scores of [CLS] token $\mathbf{Z}_{\text{cls}} \in \mathbb{R}^d$ on the other visual tokens $\mathbf{Z}_v \in \mathbb{R}^{n \times d}$ are calculated by

$$\mathbf{a}_{\text{[CLS]}} = \text{softmax}(\ \mathbf{Z}_{\text{cls}} \mathbf{W} q (\mathbf{Z}_v \mathbf{W}_k)^\top / \sqrt{d}\ ), \qquad (1)$$

Figure 4: Illustration of CondenseVLM. We first re-rank image tokens using [CLS] attention from the visual encoder and prune last $R\%$. The remaining image tokens, after passing through the multimodal projector, are combined with language instructions as input to LLMs for efficient response generation.

where $n$ represents the number of visual tokens, $d$ denotes the dimensionality of the hidden states, and $\mathbf{W}_q$ and $\mathbf{W}_k$ are the transformation matrices for the query and key projections at the current layer. The selection process begins by identifying those tokens with the highest attention score as faithful anchors. Subsequently, other pivotal tokens are selected through an iterative competitive process, which prioritizes tokens with high attention scores and discourages the selection of semantically similar tokens, for diversity within the retained set. The score of candidate tokens is calculated as

$$S_{anchor} = \frac{\mathbb{E}[|\mathbf{S}|]}{\mathbb{E}[|\mathbf{A}|]} \cdot \mathbf{A} - \max_{j \in \mathcal{T}_a} \cos(\mathbf{Z}_{v,m}, \mathbf{Z}_{v,j}), \tag{2}$$

where $\mathbf{S}$ is the cosine similarity matrix between visual tokens and $\mathbf{A}$ represents the attention matrix, $\mathbb{E}[|\cdot|]$ denotes the mean of absolute values, and $\mathcal{T}_a$ represents the set of previously selected visual anchor tokens. At each step, we select the highest-scoring visual token that has not yet been chosen. This iterative process continues until the pruning preservation ratio of $\alpha R\%$ is met, ensuring that we retain only the most influential and representative visual tokens as faithful anchors.

## 3.2 BACKGROUND TOKEN SELECTION

To preserve critical background context and mitigate information loss during merging, we introduce a specialized background token selection mechanism based on global feature coherence. This mechanism aims to retain the overall background information of images, and provide a robust fusion anchor for subsequent token merging, thereby preventing excessive redundant background information from interfering with pivotal visual anchor tokens. Specifically, we calculate the global similarity of all tokens to assess their likelihood of being background tokens. Specifically, the global score is computed through the sum of cosine similarities between each visual token and others,

$$S_{global} = \sum_{i=1}^{n} \cos(\mathbf{Z}_{v,m}, \mathbf{Z}_{v,i}), \tag{3}$$

where $\cos(\mathbf{Z}_{v,m}, \mathbf{Z}_{v,i})$ represents the cosine similarity between the $m$-th token and the $i$-th token, and $S_{global}$ is the sum of similarities for each token. Following this, we select the token with the maximum sum of similarities as the initial background token. Subsequently, for each subsequent background token selection, the score for each candidate token is calculated as follows

$$S_{background} = \frac{\mathbb{E}[|\mathbf{S}|]}{\mathbb{E}[|S_{global}|]} \cdot S_{global} - \max_{j \in \mathcal{T}_b} \cos(\mathbf{Z}_{v,m}, \mathbf{Z}_{v,j}), \tag{4}$$

where $\mathcal{T}_b$ represents the set of previously selected visual background tokens. In each step, we select the token with the highest $S_{background}$ score as a background token until a total of $(1 - \alpha)R\%$ visual tokens are selected. This process ensures the preservation of essential background tokens with textures and contextual information. The selected visual background tokens, along with the pivotal visual anchor tokens, serve as anchor points for the spatial-similarity contextual merging stage.

## 3.3 SPATIAL-SIMILARITY CONTEXTUAL MERGING

After selecting the pivotal tokens and background tokens, we perform a spatial-similarity contextual merging to integrate the remaining tokens into these representative anchors. We design a hybrid merging strategy based on similarity and spatial proximity to minimize information loss during this process. In particular, to compute the spatial distance between tokens, we map each token back to its corresponding two-dimensional coordinates on the original image grid. Given a feature map resolution of $H \times W$, each token is associated with a coordinate $(x, y)$ where $0 \le x < W$ and $0 \le y < H$. These coordinates represent the spatial location of the tokens in the original image. First, we define a remaining mask $\mathcal{M}$ to identify those not selected visual tokens as follows

$$\mathcal{M}_i = \begin{cases} \text{False}, & \text{if } i \in \mathcal{T}_a \cup \mathcal{T}_b \\ \text{True}, & \text{otherwise} \end{cases}, \tag{5}$$

where $\mathcal{T}_a$ and $\mathcal{T}_b$ are the sets of selected high-information-density and background tokens, respectively. For each remaining token $\mathbf{Z}_{v,i}$ where $\mathcal{M}_i = \text{True}$, we assign it to the nearest selected token based on a combined spatial and semantic similarity score. The assignment score $s(i, j)$ between the $i$-th remaining visual token and the $j$-th selected token is defined as:

$$S_{group}(i, j) = dist_{\cos}(\mathbf{Z}_{v,i}, \mathbf{Z}_{v,j}) - \frac{dist_{spat}(c_i, c_j)}{\max(dist_{spat}(c_i, c_j))}, \tag{6}$$

where $dist_{\cos}(\cdot, \cdot)$ denotes cosine similarity between tokens, and $dist_{spat}(c_i, c_j)$ is the spatial distance between coordinates $c_i$ and $c_j$ of $i$-th and $j$-th tokens. The assignment is determined by

$$\text{Assignment}(i) = \arg\max_{j \in \mathcal{T}_a \cup \mathcal{T}_b} S_{group}(i, j). \tag{7}$$

Once remaining tokens are assigned to clusters, we aggregate tokens within clusters, and the merged token $\mathbf{Z}_{\text{merge},k}$ for the $k$-th cluster is computed as a weighted average of all tokens within clusters,

$$\mathbf{Z}_{\text{merge},k} = \frac{\sum_{i \in \mathcal{C}_k} \mathbf{a}_{[\text{CLS}],i} \cdot \mathbf{Z}_{v,i}}{\sum_{i \in \mathcal{C}_k} \mathbf{a}_{[\text{CLS}],i}}, \tag{8}$$

where $\mathcal{C}_k$ is the set of tokens assigned to the $k$-th cluster (including the central selected token), and $\mathbf{a}_{[\text{CLS}],i}$ is the attention score of the $i$-th token. Finally, the merged visual tokens are sorted based on their original positions to maintain the spatial order of the visual features, ensuring that the downstream multimodal reasoning tasks can utilize the compressed visual representation effectively.

## 4 EXPERIMENTS

**Experiment Setting.** We conduct experiments on multiple MLLMs across diverse multimodal benchmarks. For the details on implementation and related settings, please refer to the Appendix A.

### 4.1 MAIN RESULT

The results presented in Table 2 highlight the exceptional performance of CondenseVLM, on a diverse range of image understanding tasks under varying vision token configurations. We can observe that with only 192 tokens retained ($\downarrow 66.7\%$), CondenseVLM achieves an impressive average performance of 99.5%, substantially outperforming the second-best method, MustDrop, by 2.3%. When the number of retained tokens is reduced further to 64 tokens, CondenseVLM only decreases the original performance by 3.0%. As shown in Fig. 5, compared to attention-based methods, our approach maintains substantial performance even under extreme pruning rates. In particular, as shown in Table 4, our method surpasses VisionZIP in the vast majority of benchmarks, which is based solely on attention scores to select tokens and similarity to merge tokens. In contrast, CondenseVLM achieves superior performance by strategically selecting unique tokens based on similarity, and integrating redundant tokens based on a combination of similarity and spatial distance. These results underscore CondenseVLM's effectiveness in preserving key information with a limited token budget.

Table 2: Performance comparison of various methods across different benchmarks. Results are shown for different pruning ratios, with accuracy and average performance highlighted. Best results in pink.

| Methods | GQA | MMB | MMB$_{CN}$ | MME | POPE | SQA | VQA$_{V2}$ | VQA$_{Text}$ | VizWiz | Avg. |
|---|---|---|---|---|---|---|---|---|---|---|
| Upper Bound, 576 Tokens | 61.9 | 64.7 | 58.1 | 1862 | 85.9 | 69.5 | 78.4 | 58.2 | 50.0 | 100% |
| LLaVA-1.5-7B | | | | | *Retain 192 Tokens* (↓ **66.7**%) | | | | | |
| ToMe (ICLR23) | 54.3 | 60.5 | - | 1563 | 72.4 | 65.2 | 68.0 | 52.1 | - | 88.5% |
| FastV (ECCV24) | 52.7 | 61.2 | 57.0 | 1612 | 64.8 | 67.3 | 67.1 | 52.5 | 50.8 | 90.5% |
| LLaVA-PruMerge (ICCV25) | 54.3 | 59.6 | 52.9 | 1632 | 71.3 | 67.9 | 70.6 | 54.3 | 50.1 | 91.4% |
| PDrop (CVPR25) | 57.1 | 63.2 | 56.8 | 1766 | 82.3 | 68.8 | 75.1 | 56.1 | 51.1 | 96.7% |
| FiCoCo-V (2025.03) | 58.5 | 62.3 | 55.3 | 1732 | 82.5 | 67.8 | 74.4 | 55.7 | 51.0 | 96.1% |
| MustDrop (2024.11) | 58.2 | 62.3 | 55.8 | 1787 | 82.6 | 69.2 | 76.0 | 56.5 | 51.4 | 97.2% |
| HiRED (AAAI25) | 58.7 | 62.8 | 54.7 | 1737 | 82.8 | 68.4 | 74.9 | 47.4 | 50.1 | 94.6% |
| SparseVLM (ICML25) | 57.6 | 62.5 | 53.7 | 1721 | 83.6 | 69.1 | 75.6 | 56.1 | 50.5 | 96.1% |
| DART (EMNLP25) | 58.9 | 63.6 | 57.0 | 1856 | 82.8 | 69.8 | 76.7 | 57.4 | 51.1 | 98.5% |
| CondenseVLM (Ours) | 59.8 | 65.3 | 57.3 | 1794 | 87.1 | 69.6 | 77.7 | 57.5 | 51.7 | 99.5% |
| LLaVA-1.5-7B | | | | | *Retain 128 Tokens* (↓ **77.8**%) | | | | | |
| ToMe (ICLR23) | 52.4 | 53.3 | - | 1343 | 62.8 | 59.6 | 63.0 | 49.1 | - | 80.4% |
| FastV (ECCV24) | 49.6 | 56.1 | 56.4 | 1490 | 59.6 | 60.2 | 61.8 | 50.6 | 51.3 | 85.4% |
| LLaVA-PruMerge (ICCV25) | 53.3 | 58.1 | 51.7 | 1554 | 67.2 | 67.1 | 68.8 | 54.3 | 50.3 | 89.4% |
| PDrop (CVPR25) | 56.0 | 61.1 | 56.6 | 1644 | 82.3 | 68.3 | 72.9 | 55.1 | 51.0 | 94.9% |
| FiCoCo-V (2025.03) | 57.6 | 61.1 | 54.3 | 1711 | 82.2 | 68.3 | 73.1 | 55.6 | 49.4 | 94.9% |
| MustDrop (2024.11) | 56.9 | 61.1 | 55.2 | 1745 | 78.7 | 68.5 | 74.6 | 56.3 | 52.1 | 95.7% |
| HiRED (AAAI25) | 57.2 | 61.5 | 53.6 | 1710 | 79.8 | 68.1 | 73.4 | 46.1 | 51.3 | 93.1% |
| SparseVLM (ICML25) | 56.0 | 60.0 | 51.1 | 1696 | 80.5 | 67.1 | 73.8 | 54.9 | 51.4 | 93.8% |
| DART (EMNLP25) | 57.9 | 63.2 | 57.0 | 1845 | 80.1 | 69.1 | 75.9 | 56.4 | 51.7 | 97.5% |
| CondenseVLM (Ours) | 58.9 | 64.5 | 56.3 | 1747 | 86.2 | 69.7 | 76.3 | 57.1 | 52.9 | 98.8% |
| LLaVA-1.5-7B | | | | | *Retain 64 Tokens* (↓ **88.9**%) | | | | | |
| ToMe (ICLR23) | 48.6 | 43.7 | - | 1138 | 52.5 | 50.0 | 57.1 | 45.3 | - | 70.1% |
| FastV (ECCV24) | 46.1 | 48.0 | 52.7 | 1256 | 48.0 | 51.1 | 55.0 | 47.8 | 50.8 | 76.7% |
| LLaVA-PruMerge (ICCV25) | 51.9 | 55.3 | 49.1 | 1549 | 65.3 | 68.1 | 67.4 | 54.0 | 50.1 | 87.7% |
| PDrop (CVPR25) | 41.9 | 33.3 | 50.5 | 1092 | 55.9 | 68.6 | 69.2 | 45.9 | 50.7 | 77.5% |
| FiCoCo-V (2025.03) | 52.4 | 60.3 | 53.0 | 1591 | 76.0 | 68.1 | 71.3 | 53.6 | 49.8 | 91.5% |
| MustDrop (2024.11) | 53.1 | 60.0 | 53.1 | 1612 | 68.0 | 63.4 | 69.3 | 54.2 | 51.2 | 90.1% |
| HiRED (AAAI25) | 54.6 | 60.2 | 51.4 | 1599 | 73.6 | 68.2 | 69.7 | 44.2 | 50.2 | 89.4% |
| SparseVLM (ICML25) | 52.7 | 56.2 | 46.1 | 1505 | 75.1 | 62.2 | 68.2 | 51.8 | 50.1 | 87.3% |
| DART (EMNLP25) | 55.9 | 60.6 | 53.2 | 1765 | 73.9 | 69.8 | 72.4 | 54.4 | 51.6 | 93.9% |
| CondenseVLM (Ours) | 56.5 | 63.1 | 55.1 | 1695 | 82.8 | 70.1 | 74.2 | 55.6 | 54.1 | 97.0% |

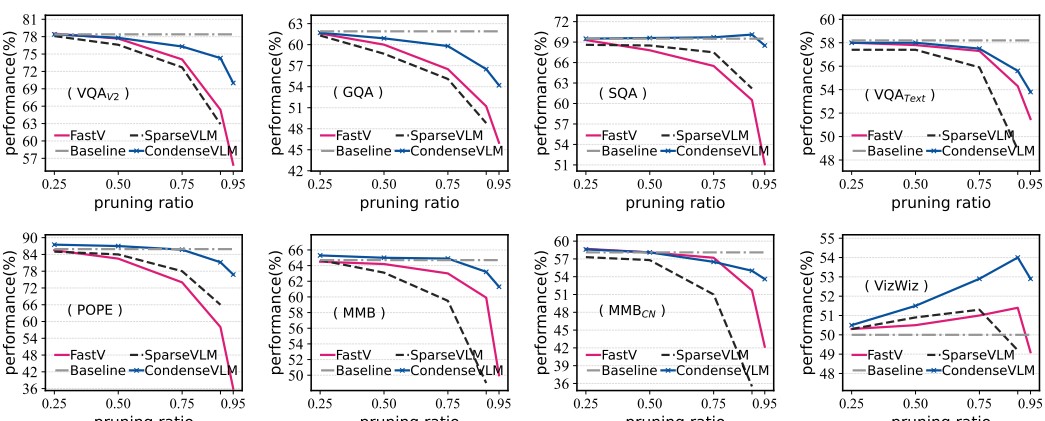

Figure 5: Relationship between performance and pruning ratios of different baseline methods. As the token pruning ratio grows, the performance of these attention-based strategies degrades dramatically, while CondenseVLM maintains the favorable performance even at 90% and 95% of pruning ratios.

## 4.2 CONDENSEVLM WITH HIGHER RESOLUTION

High-resolution image introduces many more visual tokens, further intensifying the computational load on the VLM. In this section, we apply CondenseVLM to LLaVA-NeXT-7B, which can handle up to 2880 visual tokens. The results are presented in Table 3. Compared to LLaVA-1.5, LLaVA-NeXT involves a greater number of visual tokens, implying a higher degree of redundancy. Under 88.9% pruning ratio, retaining only 320 visual tokens, CondenseVLM preserves 96.1% of the original

Table 3: Performance comparison of various methods across different benchmarks. Results are shown for different pruning ratios, with accuracy and average performance highlighted. Best results in pink.

| Methods | GQA | MMB | MMB$_{CN}$ | MME | POPE | SQA | VQA$_{V2}$ | VQA$_{Text}$ | VizWiz | Avg. |
|---|---|---|---|---|---|---|---|---|---|---|
| Upper Bound, 2880 Tokens | 64.2 | 67.4 | 60.6 | 1851 | 86.5 | 70.1 | 81.8 | 64.9 | 57.6 | 100% |
| LLaVA-NeXT-7B | | | | | *Retain 320 Tokens* (↓ **88.9%**) | | | | | |
| FastV (ECCV24) | 55.9 | 61.6 | 51.9 | 1661 | 71.7 | 62.8 | 71.9 | 55.7 | 53.1 | 88.0% |
| LLaVA-PruMerge (ICCV25) | 53.6 | 61.3 | 55.3 | 1534 | 60.8 | 66.4 | 69.7 | 50.6 | 54.0 | 85.6% |
| PDrop (CVPR25) | 56.4 | 63.4 | 56.2 | 1663 | 77.6 | 67.5 | 73.5 | 54.4 | 54.1 | 90.9% |
| MustDrop (2024.11) | 57.3 | 62.8 | 55.1 | 1641 | 82.1 | 68.0 | 73.7 | 59.9 | 54.0 | 92.2% |
| FasterVLM (ICCV25) | 56.9 | 61.6 | 53.5 | 1701 | 83.6 | 66.5 | 74.0 | 56.5 | 52.6 | 91.1% |
| HiRED (AAAI25) | 59.3 | 64.2 | 55.9 | 1690 | 83.3 | 66.7 | 75.7 | 58.8 | 54.2 | 93.3% |
| SparseVLM (ICML25) | 56.1 | 60.6 | 54.5 | 1533 | 82.4 | 66.1 | 71.5 | 58.4 | 52.0 | 89.7% |
| GlobalCom$^2$ (2025.2) | 57.1 | 61.8 | 53.4 | 1698 | 83.8 | 67.4 | 76.7 | 57.2 | 54.6 | 92.2% |
| DART (EMNLP25) | 61.7 | 65.3 | 58.2 | 1710 | 84.1 | 68.4 | 79.1 | 58.7 | 56.1 | 93.9% |
| CondenseVLM (Ours) | 61.8 | 65.1 | 57.6 | 1745 | 84.2 | 68.8 | 79.6 | 60.1 | 56.3 | 96.1% |

Table 4: Performance comparison of CondenseVLM with VisionZip at compression ratio of 88.9%.

| Methods | GQA | MMB | MMB$_{CN}$ | MME | POPE | SQA | VQA$_{V2}$ | VQA$_{Text}$ | VizWiz | Avg. |
|---|---|---|---|---|---|---|---|---|---|---|
| Upper Bound, 576 Tokens | 61.9 | 64.7 | 58.1 | 1862 | 85.9 | 69.5 | 78.4 | 58.2 | 50.0 | 100% |
| LLaVA-1.5-7B | | | | | *Retain 192 Tokens* (↓ **66.7%**) | | | | | |
| VisionZip (CVPR25) | 59.3 | 64.5 | 57.3 | 1767 | 86.4 | 68.9 | 76.8 | 57.3 | 51.6 | 98.1% |
| CondenseVLM (Ours) | 59.8 | 65.3 | 57.3 | 1794 | 87.1 | 69.6 | 77.7 | 57.5 | 51.7 | 99.5% |
| LLaVA-1.5-7B | | | | | *Retain 128 Tokens* (↓ **77.8%**) | | | | | |
| VisionZip (CVPR25) | 57.6 | 63.4 | 56.7 | 1768 | 84.7 | 68.8 | 75.6 | 56.8 | 52.0 | 97.2% |
| CondenseVLM (Ours) | 58.9 | 64.5 | 56.3 | 1747 | 86.2 | 69.7 | 76.3 | 57.1 | 52.9 | 98.8% |
| LLaVA-1.5-7B | | | | | *Retain 64 Tokens* (↓ **88.9%**) | | | | | |
| VisionZip (CVPR25) | 55.1 | 60.1 | 55.4 | 1690 | 77.0 | 69.0 | 72.4 | 55.5 | 52.9 | 94.5% |
| CondenseVLM (Ours) | 56.5 | 63.1 | 55.1 | 1695 | 82.8 | 70.1 | 74.2 | 55.6 | 54.1 | 97.0% |

performance, significantly outperforming FastV (88.0%), SparseVLM (89.7%), as well as FasterVLM (91.1%), demonstrating the robustness of CondenseVLM even to high-resolution visual inputs.

### 4.3 CONDENSEVLM WITH OTHER MLLM ARCHITECTURE

To further verify the architectural generalization of CondenseVLM beyond LLaVA families, we conduct experiments on the Qwen2.5-VL-7B (Bai et al., 2025) architecture. As shown in Tab. 6, CondenseVLM demonstrates strong generalization capability across this architecture, consistently outperforming the text-visual attention-based FastV at different pruning ratios. Of note, it achieves average performance retention proportion of 95.1%, 93.3%,

Table 6: Comparative experiments on Qwen2.5-VL-7B model.

| Methods | MMB | MME | POPE | SQA | VQA$_{Text}$ | Avg. |
|---|---|---|---|---|---|---|
| Upper Bound | 82.8 | 2304 | 86.1 | 84.7 | 84.8 | 100.% |
| Qwen2.5-VL-7B | | *Token Pruning Rate = 66.7%* | | | | |
| FastV (ECCV24) | 75.7 | 2072 | 82.2 | 78.5 | 77.9 | 92.3% |
| CondenseVLM (Ours) | 80.0 | 2133 | 83.3 | 79.9 | 78.9 | 95.1% |
| Qwen2.5-VL-7B | | *Token Pruning Rate = 77.8%* | | | | |
| FastV (ECCV24) | 74.9 | 2036 | 80.7 | 78.0 | 69.0 | 89.2% |
| CondenseVLM (Ours) | 77.0 | 2099 | 82.7 | 78.9 | 69.5 | 93.3% |
| Qwen2.5-VL-7B | | *Token Pruning Rate = 88.9%* | | | | |
| FastV (ECCV24) | 69.2 | 1940 | 78.6 | 77.4 | 60.3 | 84.3% |
| CondenseVLM (Ours) | 73.2 | 2014 | 79.7 | 78.5 | 61.2 | 90.3% |

and 90.3% at 66.7%, 77.8%, and 88.9% token pruning rates, respectively, significantly higher than FastV's 92.3%, 89.2%, and 84.3%. More results are listed on Table 9 (*in the Appendix*). Besides, we conducted supplement experiments on InternVL2 and LLaVA-OneVision-1.5, as in Table 5.

Table 5: Performance comparison on LLaVA-OneVision-1.5-8B model with 90% pruning ratios.

| Method | VizWiz | GQA | TextVQA | MME | MMB | POPE | Avg. |
|---|---|---|---|---|---|---|---|
| LLaVA-OneVision-1.5-8B | 66.0 | 69.2 | 79.5 | 2271.3 | 85.3 | 88.5 | 100% |
| FastV (ECCV24) | 60.9 (92.3%) | 61.3 (88.6%) | 56.5 (71.1%) | 1800.0 (79.2%) | 71.1 (83.4%) | 62.9 (71.1%) | 80.9% |
| PDrop (CVPR25) | 58.8 (89.1%) | 61.5 (88.9%) | 55.3 (69.6%) | 1829.7 (80.6%) | 70.5 (82.6%) | 69.9 (79.0%) | 81.6% |
| VisionZip (CVPR25) | 59.8 (90.6%) | 60.7 (87.7%) | 48.2 (60.6%) | 1980.3 (87.2%) | 73.3 (85.9%) | 79.3 (89.6%) | 83.6% |
| CondenseVLM (Ours) | 62.6 (94.8%) | 63.1 (91.2%) | 63.4 (79.7%) | 2002.4 (88.2%) | 77.6 (91.0%) | 83.9 (94.8%) | 90.0% |

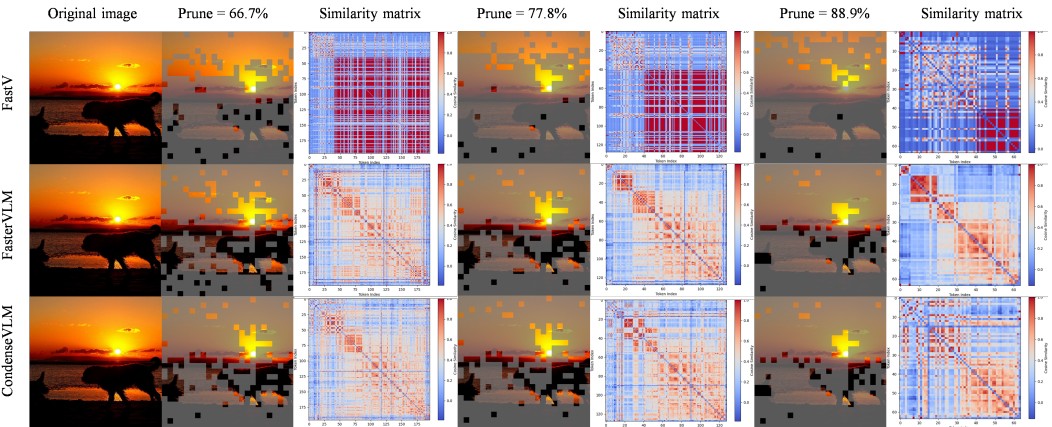

Figure 6: The case comparison between FastV, FasterVLM and CondenseVLM. It presents original images alongside their pruned versions at pruning rates of 66.7%, 77.8%, and 88.9%, along with their corresponding similarity matrices. CondenseVLM effectively preserves pivotal visual tokens.

These results demonstrate that our proposed '*Zero in on faithful anchors*' strategy in CondenseVLM effectively generalizes across different MLLM architectures (More experiments in App C.2, C.3).

### 4.4 ABLATION STUDIES

To validate the effectiveness across the factors we introduced in our CondenseVLM, we conduct ablation experiments. All ablation experiments are conducted on LLaVA-1.5-7B, and performance is assessed across benchmarks.

Table 7: Ablation study across different factors introduced in CondenseVLM's setting with the pruning ratio of 90%.

| Selection | | Merging | | MME | MMVet | POPE | VQA$_{Text}$ | Avg. |
|---|---|---|---|---|---|---|---|---|
| $\mathcal{A}_p$ | $\mathcal{S}_p$ | $\mathcal{S}_m$ | $\mathcal{D}_m$ | 1862 | 31.1 | 85.9 | 58.2 | 100.% |
| ✓ | | | | 1637 | 30.1 | 79.5 | 55.3 | 93.1% |
| ✓ | ✓ | | | 1662 | 30.7 | 81.4 | 55.9 | 84.6% |
| ✓ | ✓ | ✓ | | 1664 | 30.7 | 81.6 | 55.8 | 94.7% |
| ✓ | ✓ | ✓ | ✓ | 1695 | 31.8 | 82.0 | 55.6 | 96.2% |

***Impact of redundant tokens***. We began by investigating how redundant tokens in the token selection process influence overall performance. To do this, we compared two distinct token selection approaches, as detailed in the first two rows of Table 7. The first approach selected tokens solely based on attention scores ($\mathcal{A}_p$). The second approach, while still relying on attention scores, also actively suppressed the similarity between the selected tokens ($\mathcal{A}_p + \mathcal{S}_p$). Our findings revealed that this second approach, which prioritizes diversity among selected tokens, yielded significant performance improvements across a range of benchmarks. Further visual evidence is presented in Fig. 6, which demonstrates that CondenseVLM effectively reduces the similarity between retained tokens, and such results appear evident under higher pruning rates. As a result, redundant information restricts models' performance, and CondenseVLM effectively reduces those redundant visual tokens for efficiency.

***Impact of background tokens***. We then explored the influence of the number of background tokens on performance (Fig. 7). Neither the absence nor the existence of too many background tokens led to suboptimal performance. Upon excluding background tokens, the model relies on [CLS] attention, which prevents a full representation of the feature space.. Conversely, an excessive number of background tokens can dilute the model's focus due to the inclusion of less relevant visual tokens.

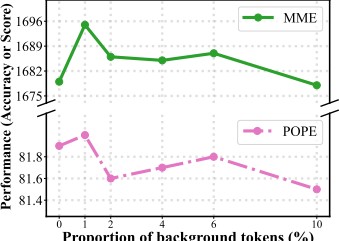

Figure 7: Impact of the proportion of background tokens. Experiments are conducted on LLaVA-1.5-7B with the pruning ratio of 90%.

***Impact of Spatial Distance Constraints***. Finally, we investigated the impact of incorporating spatial distance constraints during the token merging phase of our method. As shown in the last two rows of Table 7, we specifically compared two token merging strategies: one based solely on similarity ($\mathcal{S}_m$), and another that incorporates both similarity and spatial distance ($\mathcal{S}_m + \mathcal{D}_m$). Our analysis revealed that integrating spatial distance into the similarity-based merging strategy led to enhanced performance across most scene understanding benchmarks. This finding indicates that a merging approach accounting for both similarity and spatial distance effectively alleviates the erroneous fusion of tokens representing distinct semantic meanings within

Table 8: Real inference comparison on POPE. Experiments adopt 66.7% and 90% pruning ratios.

| Methods | Time | Prefill | Latency | Mem. | Acc. | Time | Prefill | Latency | Mem. | Acc. |
|---|---|---|---|---|---|---|---|---|---|---|
| Upper Bound, 576 Tokens | 49:41 | 0.5ms | 0.334s | 19.0G | 100.% | 49:41 | 0.5ms | 0.334s | 19.0G | 100.% |
| LLaVA-1.5-7B | *Retain 192 Tokens* (↓ **66.7**%) | | | | | *Retain 58 Tokens* (↓ **90**%) | | | | |
| FastV (ECCV24) | 35:34 | 0.5ms | 0.239s | 16.0G | 75.4% | 30:41 | 0.5ms | 0.206s | 15.6G | 66.8% |
| MustDrop (2024.11) | 32:30 | 0.5ms | 0.273s | 15.6G | 96.2% | 29:40 | 0.6ms | 0.199s | 14.5G | 87.1% |
| FasterVLM (ICCV25) | 30:09 | 0.5ms | 0.202s | 15.6G | 100.% | 25:08 | 0.5ms | 0.168s | 14.5G | 92.5% |
| HiRED (AAAI25) | 30:08 | 0.6ms | 0.210s | 15.7G | 96.4% | 25:03 | 0.6ms | 0.168s | 14.5G | 92.7% |
| SparseVLM (ICML25) | 40:51 | 0.6ms | 0.251s | 15.8G | 97.3% | 31:28 | 0.6ms | 0.212s | 14.6G | 92.3% |
| CondenseVLM (Ours) | 31:02 | 0.5ms | 0.208s | 15.6G | 101.% | 28:28 | 0.5ms | 0.191s | 14.5G | 95.5% |

a scene. Notably, however, our token merging strategy delivered suboptimal results on image-text understanding tasks such as TextVQA. This phenomenon may arise from erroneous merging, where text-related tokens incorrectly fuse with surrounding contextual tokens, distorting the final outcome, which in turn has further inspired us to refine our methodology.

## 4.5 EFFICIENCY ANALYSIS

To demonstrate the efficiency of CondenseVLM, we compared the total inference time, prefill time, latency, GPU memory consumption, and accuracy of different methods on LLaVA-1.5-7B. As illustrated in Table 8, under pruning ratios of 66.7% and 90%, CondenseVLM achieves a 42.7% reduction in total inference time and a 42.8% decrease in latency, with only a minimal 4.5% drop in accuracy. Compared to FastV, our proposed method not only consumes less GPU memory but also delivers faster inference speed. Although our total inference time is slightly slower than that of FasterVLM, CondenseVLM outperforms FasterVLM by 3.0% in accuracy, successfully establishing an optimal balance between computational efficiency and model performance.

## 5 RELATED WORK

### 5.1 TOKEN COMPRESSION FOR MLLMS

The inclusion of visual information in MLLMs introduces long token sequences, leading to high computation and memory costs. For example, mini-Gemini-HD (Li et al., 2024c) generates 2880 tokens from high-definition images, creating inference bottlenecks. To address this, research has focused on token compression and pruning techniques in Vision Transformers (Bolya et al., 2022) and MLLMs (Huang et al., 2024a). Methods like LLaMA-VID (Li et al., 2024b) and DeCo (Yao et al., 2024) address this by modifying models and adding training, which increases computational costs. ToMe (Bolya et al., 2022) reduces tokens without training but disrupts early cross-modal interactions (Xing et al., 2024). LLaVA-PruMerge (Shang et al., 2024) selectively retains key tokens while merging less critical ones based on key similarity. FasterVLM (Zhang et al., 2024a) utilizes [CLS] attention scores from the visual encoder to re-rank and retain top visual tokens. FastV (Chen et al., 2024) and SparseVLM (Zhang et al., 2024b) focus on token selection using attention scores or cross-modal guidance, but overlook the role of token duplication and lack Flash-Attention (Dao et al., 2022; Dao, 2024). CondenseVLM maintains hard acceleration compatibility (*e.g.*, Flash-Attention), and effectively retains visual context during aggressive pruning. More works are listed in Appx. B.

## 6 CONCLUSION

We present CondenseVLM, a plug-and-play framework that enhances MLLM efficiency by aggressively reducing visual tokens while preserving critical visual context. Unlike prior approaches that rely solely on attention, CondenseVLM combines attention-based selection, similarity suppression, background compensation, and spatial-semantic merging. This ensures semantic diversity and contextual integrity. Experiments show that CondenseVLM prunes up to 77.8% of tokens with minimal performance drop, offering significant speed and memory gains. Our results underline the value of jointly considering attention, redundancy, and spatial cues for scalable vision-language modeling.

*We illustrate the broader impact of CondenseVLM and LLM usage in Section I and J, respectively.*

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

## ∞ Technical Appendices and Supplements

In this appendix, we provide detailed information regarding the experimental setup, encompassing the datasets, model architectures, and comparison methods. Furthermore, we present additional visualizations and insights concerning the redundancy within tokens selected by attention-based methods, the distribution of [CLS] attention, and the similarity patterns between background tokens and other tokens. We also present fine-grained results on the MMBench, and the use of LLMs.

## A DETAILED EXPERIMENT SETTINGS

### A.1 DATASETS

We conducted experiments on several widely used visual understanding benchmarks. For image understanding task, we performed experiments on ten widely used benchmarks, including GQA (Hudson & Manning, 2019), MMBench (MMB) and MMB-CN (Liu et al., 2025b), MME (Fu et al., 2023), POPE (Li et al., 2023), VizWiz (Bigham et al., 2010), SQA (ScienceQA) (Lu et al., 2022), VQA$_{V2}$ (VQA V2) (Goyal et al., 2017), VQA$_{Text}$ (TextVQA) (Singh et al., 2019) and MMVet (Yu et al., 2023)

**GQA.** (Hudson & Manning, 2019) The GQA benchmark is composed of three parts: scene graphs, questions, and images. The image part contains images, as well as the spatial features of images and the features of all objects in images. The questions in GQA are designed to test the understanding of visual scenes and the ability to reason about different aspects of an image.

**MMBench.** (Liu et al., 2025b) The MMBench benchmark comprehensively evaluates the model's overall performance across multiple dimensions. It includes three levels of ability dimensions. The first level (L-1) consists of two main abilities, perception and reasoning. The second level (L-2) expands based on the first level, including six sub-abilities. The third level (L-3) further refines the second level, encompassing 20 specific ability dimensions. This hierarchical structure enables a granular and comprehensive evaluation of the model's various capabilities.

**MME.** (Fu et al., 2023) The MME benchmark is also a comprehensive benchmark meticulously designed to thoroughly evaluate various aspects of a model's performance. It consists of 14 subtasks that specifically aim to evaluate both the model's perceptual and cognitive abilities. By utilizing manually constructed instruction-answer pairs and concise instruction design, it effectively mitigates issues such as data leakage and unfair evaluation of model performance.

**POPE.** (Li et al., 2023) The POPE benchmark is primarily used to evaluate the degree of Object Hallucination in models. It reformulates hallucination evaluation by requiring the model to answer a series of specific binary questions regarding the presence of objects in images. Accuracy, Recall, Precision, and F1 Score are effectively employed as reliable evaluation metrics to precisely measure the model's hallucination level under three different sampling strategies.

**ScienceQA.** (Lu et al., 2022) The ScienceQA benchmark covers a rich diversity of domains, including natural science, language science, and social science. Within each subject, questions are categorized first by the topic, then by the category, and finally by the skill. This hierarchical categorization results in 26 topics, 127 categories, and 379 skills, providing a comprehensive and diverse range of scientific questions. It provides a comprehensive evaluation of a model's capabilities in multimodal understanding, multi-step reasoning, and interpretability.

**VQA-v2.** (Goyal et al., 2017) The VQA-v2 benchmark evaluates the model's visual perception capabilities through open-ended questions. It consists of 265,016 images, covering a wide variety of real-world scenes and objects, providing rich visual contexts for the questions. For each question, there are 10 ground truth answers provided by human annotators, which allows for a comprehensive evaluation of the performance of different models in answering the questions accurately.

**TextVQA.** (Singh et al., 2019) The TextVQA benchmark focuses on the comprehensive integration of diverse text information within images. It meticulously evaluates the model's text understanding and reasoning abilities through a series of visual question-answering tasks with rich textual information. Models need to not only understand the visual content of the images but also be able to read and reason about the text within the images to answer the questions accurately.

**MMVet.** (Yu et al., 2023) The MMVet benchmark is designed based on the insight that the intriguing ability to solve complicated tasks is often achieved by a generalist model being able to integrate different core vision-language capabilities. MM-Vet defines 6 core VL capabilities and examines the 16 integrations of interest derived from the capability combination.

## A.2 MODELS

We evaluate CondenseVLM using various open-source MLLMs. For image understanding tasks, experiments are conducted on the LLaVA family, including LLaVA-1.5-7B[1] (Liu et al., 2024a) and LLaVA-Next-7B[2] (Liu et al., 2024b), with the latter used to validate performance on high-resolution images.

## A.3 BASELINES

We analyze multiple representative methods for accelerating multi-modal language models (MLLMs) through token reduction. These methods share the goal of improving efficiency by reducing redundant tokens, yet differ in their strategies, such as token merging, pruning, or adaptive allocation.

**ToMe** (Bolya et al., 2022) merges similar tokens in visual transformer layers through lightweight matching techniques, achieving acceleration without requiring additional training.

**FastV** (Chen et al., 2024) focuses on early-stage token pruning by leveraging attention maps, effectively reducing computational overhead in the initial layers.

**SparseVLM** (Zhang et al., 2024b) ranks token importance using cross-modal attention and introduces adaptive sparsity ratios, complemented by a novel token recycling mechanism.

**HiRED** (Arif et al., 2024) allocates token budgets across image partitions based on CLS token attention, followed by the selection of the most informative tokens within each partition, ensuring spatially aware token reduction.

**LLaVA-PruMerge** (Shang et al., 2024) combines pruning and merging strategies by dynamically removing less important tokens using sparse CLS-visual attention and clustering retained tokens based on key similarity.

**PDrop** (Xing et al., 2024) adopts a progressive token-dropping strategy across model stages, forming a pyramid-like token structure that balances efficiency and performance.

**MustDrop** (Liu et al., 2024d) integrates multiple strategies, including spatial merging, text-guided pruning, and output-aware cache policies, to reduce tokens across various stages.

**FasterVLM** (Zhang et al., 2024a) evaluates token importance via CLS attention in the encoder and performs pruning before interaction with the language model, streamlining the overall process.

**GlobalCom$^2$** (Liu et al., 2025a) introduces a hierarchical approach by coordinating thumbnail tokens to allocate retention ratios for high-resolution crops while preserving local details.

**VisionZip** (Yang et al., 2024a) evaluates token importance via attention in the encoder and clustering retained tokens based on key similarity.

**DART** (Wen et al., 2025) introduces a duplication-aware token reduction method that selects a small subset of pivot tokens, calculates cosine similarity between pivot tokens and remaining tokens, retains those with the lowest duplication to pivots, achieving significant acceleration while maintaining performance and good compatibility with efficient attention operators.

These methods collectively highlight diverse approaches to token reduction, ranging from attention-based pruning to adaptive merging, offering complementary solutions for accelerating MLLMs.

---

[1] https://huggingface.co/liuhaotian/llava-v1.5-7b
[2] https://huggingface.co/liuhaotian/llava-v1.6-vicuna-7b

A.4 REPRODICIBILITY STATEMENT

All of our experiments are conducted on Nvidia A800-80G GPU. The implementation was carried out in Python 3.10, utilizing PyTorch 2.1.2, and CUDA 11.8. All baseline settings follow the original paper. In Fig. 1, we define tokens as similar if their cosine similarity is greater than or equal to 0.75 (i.e., $\gamma = 0.75$). For SparseVLM, we cloned its GitHub repository on 2025.2.10, and the results of our table is real experimental results, and if you contrast SparseVLM's arxiv version, you can find its experimental results have changed from its v1v2 to v3v4 arxiv version, but its code repository has not been updated. Thus, our running results of SparseVLM are inconsistent with the 2025.02 version.

# B MORE RELATED WORK

## B.1 MLLMS AND THEIR CHALLENGES

The recent remarkable success of LLMs (Ouyang et al., 2022; Zhang et al., 2022; Touvron et al., 2023a; Dubey et al., 2024; Luo et al., 2025) has spurred the trend of applying their strong capabilities to multimodal comprehension tasks, fostering the development of MLLMs (Achiam et al., 2023; Team et al., 2023). Leveraging open-source LLMs such as LLaMA families (Touvron et al., 2023a;b; Dubey et al., 2024), MLLMs (Bai et al., 2023; Liu et al., 2024a;b) have demonstrated enhanced adaptability across a range of visual understanding tasks, leading to a more profound ability to interpret the world. While this empowers LLMs with the capability of visual perception, the incorporation of lengthy visual tokens significantly escalates the computational burdens. Moreover, studies have shown that existing MLLMs still suffer from certain visual deficiencies (Tong et al., 2024; Jiang et al., 2024b) and some hallucinations (Huang et al., 2024b; 2025). Some work mitigates these issues by increasing the resolution of input images or videos (Luo et al., 2024; Xu et al., 2024b), but this further exacerbates the computational overhead. For example, LLaVA-1.5 (Liu et al., 2024c) encodes a 336-resolution image into 576 visual tokens, while LLaVA-NeXT (Liu et al., 2024b) doubles the resolution and generates 2,880 tokens. LLaVA-OneVision (Li et al., 2024a) represents an image using 7,290 visual tokens, and Video-LLaVA (Lin et al., 2023) faces even higher costs, as it must process numerous visual tokens from multiple frames during inference. These visual tokens occupy a large portion of the context window of their LLMs. In this work, we conducted experiments on these representative models to verify CondenseVLM's applicability.

## B.2 VISUAL REDUNDANCY IDENTIFICATION

In MLLMs, visual redundancy identification facilitates the distillation of visual tokens with high informativeness for faster inference. There are two main research directions: a) Vision-centric strategies analyze the image's structure and feature distribution to discard less relevant visual tokens (Chen et al., 2024; Wang et al., 2024a). Existing approaches include spatial-similarity clustering (e.g., TokenLearner (Ryoo et al., 2021)), dynamic pruning based on attention scores (Han et al., 2024; Yang et al., 2024b; Xu et al., 2024a), and using information bottleneck or entropy metrics during the prefilling stage to estimate background redundancy. b) Instruction-centric strategies typically use cross-modal attention analysis or gradient accumulation to identify redundant tokens (Liu et al., 2024d; Zhu et al., 2024; Song et al., 2024). Tokens with low attention or negligible gradient impact are deemed redundant (He et al., 2024). Building on this, some studies explore learned importance scoring, training a lightweight end-to-end model to predict each patch's "instruction relevance," enabling even finer-grained pruning (Jiang et al., 2024a; Tu et al., 2024; Ye et al., 2025). As the existence of language bias in LLM may cause hallucinations, we use a vision-centric scheme.

## B.3 TOKEN COMPRESSION FOR MLLMS

The inclusion of visual information in MLLMs introduces long token sequences, leading to high computation and memory costs. For example, mini-Gemini-HD (Li et al., 2024c) generates 2880 tokens from high-definition images, creating inference bottlenecks. To address this, research has focused on token compression and pruning techniques in Vision Transformers (Bolya et al., 2022) and MLLMs (Huang et al., 2024a). Methods like LLaMA-VID (Li et al., 2024b) and DeCo (Yao et al., 2024) address this by modifying models and adding training, which increases computational costs. ToMe (Bolya et al., 2022) reduces tokens without training but disrupts early cross-modal

interactions (Xing et al., 2024). LLaVA-PruMerge (Shang et al., 2024) selectively retains key tokens while merging less critical ones based on key similarity. FasterVLM (Zhang et al., 2024a) utilizes [CLS] attention scores from the visual encoder to re-rank and retain top visual tokens. FastV (Chen et al., 2024) and SparseVLM (Zhang et al., 2024b) focus on token selection using attention scores or cross-modal guidance, but overlook the role of token duplication and lack Flash-Attention (Dao et al., 2022; Dao, 2024). Our proposed CondenseVLM maintains hard acceleration compatibility (*e.g.*, Flash-Attention), and effectively retains visual context during aggressive pruning.

## C  MORE EXPERIENT RESULTS

### C.1  MORE EXPERIMENTS ON QWEN2.5-VL MODELS

We supplemented extra experiments on Qwen2.5-VL, as shown in Tables 9. Our method achieves the best performance across all pruning ratios compared with other state-of-the-art methods.

Table 9: Performance comparison on Qwen2.5-VL-7B model with different token pruning ratios.

| Methods | GQA | MMB | MME | POPE | SQA | VQA$_{V2}$ | VQA$_{Text}$ | VizWiz | Avg. |
|---|---|---|---|---|---|---|---|---|---|
| Upper Bound | 61.9 | 64.7 | 1862 | 85.9 | 69.5 | 78.4 | 58.2 | 50.0 | 100% |
| Qwen2.5-VL-7B | | | | *Retain 192 Tokens* (↓ **66.7%**) | | | | | |
| FastV (ECCV24) | 61.0 (93.6%) | 75.7 (91.4%) | 2072 (89.9%) | 82.2 (95.5%) | 78.5 (92.7%) | 86.5 (93.7%) | 77.9 (91.9%) | 64.1 (93.9%) | 92.8% |
| PDrop (CVPR25) | 60.7 (93.1%) | 75.5 (91.2%) | 2043 (88.7%) | 81.8 (95.0%) | 78.0 (92.1%) | 86.6 (93.8%) | 77.2 (91.0%) | 63.7 (93.3%) | 92.3% |
| VisionZip (CVPR25) | 62.5 (95.9%) | 76.0 (91.8%) | 2097 (91.0%) | 82.9 (96.3%) | 78.8 (93.0%) | 87.0 (94.3%) | 78.3 (92.3%) | 65.0 (95.2%) | 93.7% |
| DART (EMNLP25) | 63.2 (96.9%) | 77.5 (93.6%) | 2106 (91.4%) | 83.1 (96.5%) | 77.6 (91.6%) | 85.9 (93.1%) | 78.6 (92.7%) | 65.4 (95.8%) | 93.9% |
| CondenseVLM (Ours) | 63.8 (97.9%) | 80.0 (96.6%) | 2133 (92.6%) | 83.3 (96.7%) | 79.9 (94.3%) | 87.5 (94.8%) | 78.9 (93.0%) | 65.0 (95.2%) | 95.1% |
| Qwen2.5-VL-7B | | | | *Retain 192 Tokens* (↓ **66.7%**) | | | | | |
| FastV (ECCV24) | 60.5 (92.8%) | 74.9 (90.5%) | 2036 (88.4%) | 80.7 (93.7%) | 78.0 (92.1%) | 82.3 (89.2%) | 69.5 (82.0%) | 63.5 (93.0%) | 90.2% |
| PDrop (CVPR25) | 60.2 (92.3%) | 75.0 (90.6%) | 2017 (87.5%) | 80.4 (93.4%) | 77.5 (91.5%) | 82.1 (88.9%) | 69.2 (81.6%) | 64.0 (91.1%) | 89.6% |
| VisionZip (CVPR25) | 61.8 (94.8%) | 75.7 (91.4%) | 2109 (91.5%) | 81.2 (94.3%) | 78.2 (92.3%) | 83.0 (89.9%) | 70.7 (83.4%) | 65.0 (93.7%) | 91.4% |
| DART (EMNLP25) | 62.0 (95.1%) | 76.1 (91.9%) | 2125 (92.2%) | 81.9 (95.1%) | 78.1 (92.1%) | 83.2 (90.1%) | 71.2 (84.0%) | 63.5 (93.0%) | 91.7% |
| CondenseVLM (Ours) | 62.5 (95.9%) | 77.0 (93.0%) | 2099 (91.1%) | 82.7 (96.1%) | 78.9 (93.2%) | 85.6 (92.7%) | 70.7 (83.4%) | 64.2 (94.0%) | 92.4% |
| Qwen2.5-VL-7B | | | | *Retain 192 Tokens* (↓ **66.7%**) | | | | | |
| FastV (ECCV24) | 57.2 (87.7%) | 71.2 (86.0%) | 1949 (84.6%) | 78.6 (91.3%) | 77.4 (91.4%) | 81.0 (87.8%) | 60.3 (71.1%) | 60.5 (88.6%) | 86.1% |
| PDrop (CVPR25) | 56.3 86.3% | 71.4 86.2% | 1920 83.3% | 77.0 89.4% | 76.9 90.8% | 81.5 88.3% | 60.5 71.3% | 60.3 88.3% | 85.5% |
| VisionZip (CVPR25) | 58.0 89.0% | 72.7 87.8% | 2006 87.1% | 77.5 90.0% | 77.8 91.9% | 81.7 88.5% | 61.9 73.0% | 61.5 90.0% | 87.2% |
| DART (EMNLP25) | 58.5 89.7% | 71.9 86.8% | 2042 88.6% | 77.9 90.5% | 76.9 90.8% | 81.3 88.1% | 61.7 72.8% | 61.2 89.6% | 87.1% |
| CondenseVLM (Ours) | 60.4 92.6% | 73.2 88.4% | 2014 87.4% | 79.7 92.6% | 78.5 92.7% | 83.2 90.1% | 61.3 72.3% | 62.7 91.8% | 88.5% |

### C.2  EXPERIMENTS ON INTERNVL2 MODELS

Table 10: Performance comparison on InternVL2-2B model with 50% token pruning ratio.

| Method | Pruning | VizWiz | GQA | TextVQA | MME | MMB | MM-Vet | POPE | Average |
|---|---|---|---|---|---|---|---|---|---|
| InternVL2-2B | Retain 100% Tokens | 29.4 | 57.3 | 72.0 | 1821.7 | 72.5 | 39.6 | 85.4 | 100% |
| +FastV (ECCV24) | Retain 50% Tokens | 27.1 (92.2%) | 55.8 (97.4%) | 70.7 (98.2%) | 1774.2 (97.4%) | 71.8 (99.0%) | 34.1 (86.1%) | 84.8 (99.3%) | 95.7% |
| +PDrop (CVPR25) | Retain 50% Tokens | 26.9 (91.5%) | 55.3 (96.5%) | 70.2 (98.2%) | 1800.1 (98.8%) | 71.5 (98.6%) | 33.9 (85.6%) | 82.1 (96.1%) | 95.1% |
| +VisionZip (CVPR25) | Retain 50% Tokens | 27.3 (92.9%) | 56.2 (98.1%) | 71.0 (98.2%) | 1813.0 (99.5%) | 72.1 (99.4%) | 34.6 (87.4%) | 82.9 (97.1%) | 96.1% |
| +CondenseVLM (ours) | Retain 50% Tokens | 29.0 (98.6%) | 56.8 (99.1%) | 71.8 (98.2%) | 1829.7 (100.4%) | 72.3 (99.7%) | 38.6 (97.5%) | 84.5 (98.9%) | 98.9% |

Table 11: Performance comparison on InternVL2-8B model with 50% token pruning ratio.

| Method | Pruning | VizWiz | GQA | TextVQA | MME | MMB | MM-Vet | POPE | Average |
|---|---|---|---|---|---|---|---|---|---|
| InternVL2-8B | Retain 100% Tokens | 32.9 | 62.7 | 76.6 | 2205.3 | 81.8 | 60.0 | 86.7 | 100% |
| +FastV (ECCV24) | Retain 50% Tokens | 30.3 (92.1%) | 62.0 (98.9%) | 75.6 (98.7%) | 2214.2 (100.4%) | 81.2 (99.3%) | 56.6 (94.3%) | 86.5 (99.8%) | 97.6% |
| +PDrop (CVPR25) | Retain 50% Tokens | 29.9 (90.9%) | 62.3 (99.4%) | 75.6 (98.7%) | 2193.1 (99.4%) | 81.4 (99.5%) | 56.3 (93.8%) | 86.7 (100.0%) | 97.4% |
| +VisionZip (CVPR25) | Retain 50% Tokens | 31.2 (94.8%) | 62.1 (99.0%) | 75.9 (99.1%) | 2185.4 (99.1%) | 81.3 (99.4%) | 56.8 (94.7%) | 86.8 (100.1%) | 98.0% |
| +CondenseVLM (ours) | Retain 50% Tokens | 32.1 (97.6%) | 62.3 (99.4%) | 76.2 (99.5%) | 2200.5 (99.8%) | 81.5 (99.6%) | 56.9 (94.8%) | 86.2 (99.4%) | 98.6% |

As shown in Table 10 and Table 11, our method achieves the best performance compared with FastV and PDrop on these benchmarks.

## C.3 Experiments on LLaVA-OneVision-1.5 Models

Table 12: Performance comparison on LLaVA-OneVision-1.5-8B model with different token pruning ratios.

| Method | VizWiz | GQA | TextVQA | MME | MMB | POPE | Average |
|---|---|---|---|---|---|---|---|
| **70% Token Pruning (Retain 30% Tokens)** | | | | | | | |
| LLaVA-OneVision-1.5-8B | 66.0 | 69.2 | 79.5 | 2271.3 | 85.3 | 88.5 | 100% |
| +FastV | 64.1 (97.1%) | 65.2 (94.2%) | 72.3 (90.9%) | 2019.5 (88.9%) | 79.6 (93.3%) | 70.4 (79.5%) | 90.7% |
| +PDrop | 62.5 (94.7%) | 64.0 (92.5%) | 70.2 (88.3%) | 1989.2 (87.6%) | 71.5 (83.8%) | 82.1 (92.8%) | 89.9% |
| +VisionZip | 64.7 (98.0%) | 65.9 (95.2%) | 73.2 (92.1%) | 2104.6 (92.7%) | 83.3 (97.7%) | 87.1 (98.4%) | 95.7% |
| +CondenseVLM (ours) | **65.0 (98.5%)** | **66.7 (96.4%)** | **77.1 (97.0%)** | **2210.4 (97.3%)** | **83.4 (97.8%)** | **85.7 (96.8%)** | **97.3%** |
| **90% Token Pruning (Retain 10% Tokens)** | | | | | | | |
| LLaVA-OneVision-1.5-8B | 66.0 | 69.2 | 79.5 | 2271.3 | 85.3 | 88.5 | 100% |
| +FastV | 60.9 (92.3%) | 61.3 (88.6%) | 56.5 (71.1%) | 1800.0 (79.2%) | 71.1 (83.4%) | 62.9 (71.1%) | 80.9% |
| +PDrop | 58.8 (89.1%) | 61.5 (88.9%) | 55.3 (69.6%) | 1829.7 (80.6%) | 70.5 (82.6%) | 69.9 (79.0%) | 81.6% |
| +VisionZip | 59.8 (90.6%) | 60.7 (87.7%) | 48.2 (60.6%) | 1980.3 (87.2%) | 73.3 (85.9%) | 79.3 (89.6%) | 83.6% |
| +CondenseVLM (ours) | **62.6 (94.8%)** | **63.1 (91.2%)** | **63.4 (79.7%)** | **2002.4 (88.2%)** | **77.6 (91.0%)** | **83.9 (94.8%)** | **90.0%** |

As shown in Table 12, our method achieves the best performance compared with different methods on these benchmarks.

## C.4 Performance on Text-dense Benchmarks

Table 13: Performance comparison on text-dense benchmarks with different token pruning ratios.

| Method | OCRBench | ChartQA | ChineseOCRbench | Average |
|---|---|---|---|---|
| **66.7% Token Pruning (Retain 33.3% Tokens)** | | | | |
| LLaVA1.5-7B | 297 | 224 | 24 | 100% |
| +FastV | 190 (64.0%) | 202 (90.2%) | 23 (95.8%) | 83.3% |
| +PDrop | 290 (97.6%) | 209 (93.3%) | 24 (100.0%) | 97.0% |
| +DART | 296 (99.7%) | 215 (96.0%) | 26 (108.3%) | 101.3% |
| +CondenseVLM (ours) | **296 (99.7%)** | **213 (95.1%)** | **26 (108.3%)** | **101.0%** |
| **77.8% Token Pruning (Retain 22.2% Tokens)** | | | | |
| LLaVA1.5-7B | 297 | 224 | 24 | 100% |
| +FastV | 191 (64.3%) | 183 (81.7%) | 22 (91.7%) | 79.2% |
| +PDrop | 287 (96.6%) | 190 (84.8%) | 24 (95.8%) | 92.4% |
| +DART | 296 (99.7%) | 208 (92.9%) | 25 (104.2%) | 98.9% |
| +CondenseVLM (ours) | **294 (99.0%)** | **208 (92.9%)** | **25 (104.2%)** | **98.7%** |
| **88.9% Token Pruning (Retain 11.1% Tokens)** | | | | |
| LLaVA1.5-7B | 297 | 224 | 24 | 100% |
| +FastV | 191 (64.3%) | 155 (69.2%) | 15 (62.5%) | 65.3% |
| +PDrop | 250 (84.2%) | 162 (72.3%) | 17 (70.8%) | 76.8% |
| +DART | 270 (90.9%) | 171 (76.3%) | 18 (75.0%) | 80.7% |
| +CondenseVLM (ours) | **289 (97.3%)** | **170 (75.9%)** | **17 (70.8%)** | **81.3%** |

As shown in Table 13, we added experiments on text-dense benchmarks. These results indicate that CondenseVLM effectively preserves performance even when vision tokens are highly critical for text recognition tasks.

## D Computational Complexity

To evaluate the computational complexity of MLLMs, it is essential to analyze their core components, including the self-attention mechanism and the feed-forward network (FFN). The total floating-point operations (FLOPs) required can be expressed as:

$$\text{Total FLOPs} = T \times (4Nd^2 + 2N^2d + 2Ndm), \tag{9}$$

where $T$ denotes the number of transformer layers, $N$ is the sequence length, $d$ represents the hidden dimension size, and $m$ is the intermediate size of the FFN. This equation highlights the significant

Table 14: Fine-grained comparison on MMBench Liu et al. (2025b) between FasterVLM and CondenseVLM at high pruning ratios.

| Category (dev) | Vanilla (576) | FasterVLM (↓88.9%) | Condense-VLM (↓88.9%) | FasterVLM (↓75%) | Condense-VLM (↓75%) |
|---|---|---|---|---|---|
| Action Recognition | 90.7 | 83.3 | 90.9 | 85.2 | 92.0 |
| Attribute Comparison | 50.0 | 56.8 | 52.6 | 56.8 | 51.3 |
| Attribute Recognition | 79.7 | 78.4 | 74.7 | 78.4 | 78.8 |
| Celebrity Recognition | 79.8 | 82.8 | 75.4 | 81.2 | 73.7 |
| Function Reasoning | 75.9 | 77.2 | 81.1 | 75.9 | 82.1 |
| Future Prediction | 45.0 | 37.5 | 55.6 | 37.5 | 58.3 |
| Identity Reasoning | 93.3 | 93.3 | 96.3 | 95.6 | 97.6 |
| Image Emotion | 78.0 | 78.0 | 68.7 | 80.0 | 72.3 |
| Image Quality | 35.8 | 30.2 | 38.8 | 39.6 | 40.0 |
| Image Scene | 96.2 | 93.3 | 92.2 | 96.2 | 90.7 |
| Image Style | 77.4 | 66.0 | 62.4 | 69.8 | 68.2 |
| Image Topic | 83.3 | 86.1 | 94.1 | 83.3 | 95.3 |
| Nature Relation | 41.7 | 33.3 | 55.6 | 35.4 | 54.3 |
| Object Localization | 39.5 | 34.6 | 26.0 | 35.8 | 28.8 |
| OCR | 59.0 | 53.8 | 81.8 | 53.8 | 80.5 |
| Physical Property Reasoning | 50.7 | 50.7 | 55.0 | 50.7 | 57.0 |
| Physical Relation | 33.3 | 20.8 | 28.8 | 20.8 | 26.9 |
| Social Relation | 88.4 | 67.4 | 67.9 | 67.4 | 78.2 |
| Spatial Relationship | 17.8 | 20.0 | 17.3 | 22.2 | 21.0 |
| Structured Image-Text Understanding | 26.9 | 26.9 | 21.8 | 26.9 | 21.8 |

impact of sequence length $N$ on computational complexity. Notable, we follow FastV (Chen et al., 2024) to roughly estimate various token reduction baseline FLOPs. The FLOPs after token pruning can be represented as:

$$
\begin{aligned}
\text{Post-Pruning FLOPs} \\
= L \times (4Nd^2 + 2N^2d + 2Ndm) + \\
(T - L) \times (4\hat{N}d^2 + 2\hat{N}^2d + 2\hat{N}dm),
\end{aligned} \tag{10}
$$

where $L$ denotes the pruned layer, $\hat{N}$ represents token sequence length after pruning. The theoretical FLOPs reduction ratio related to visual tokens is computed as:

$$
1 - \frac{\text{Post-Pruning FLOPs}}{\text{Total FLOPs}}. \tag{11}
$$

# E    MORE VISUALIZATION ABOUT CONDENSEVLM

## E.1    SPARSIFICATION VISUALIZATION

Figure 8 presents visualizations of the remaining tokens for different methods under varying pruning ratios. We observe that FastV, due to the positional bias inherent in its LLM's text-visual attention mechanism, exhibits significant redundancy among its retained tokens. In contrast, the [CLS] attention-driven token selection approach (FasterVLM) demonstrates less redundancy and a more concentrated token distribution. This further indicates that relying solely on [CLS] attention for token selection leads to a localized clustering effect, hindering the retention of a comprehensive and representative set of key tokens across the entire image. This observation is consistent with the [CLS] attention distribution depicted in Figure 9. Our method, by limiting the similarity between selected tokens, effectively reduces redundancy within the retained token set, enabling a more comprehensive preservation of key tokens from diverse locations within the image.

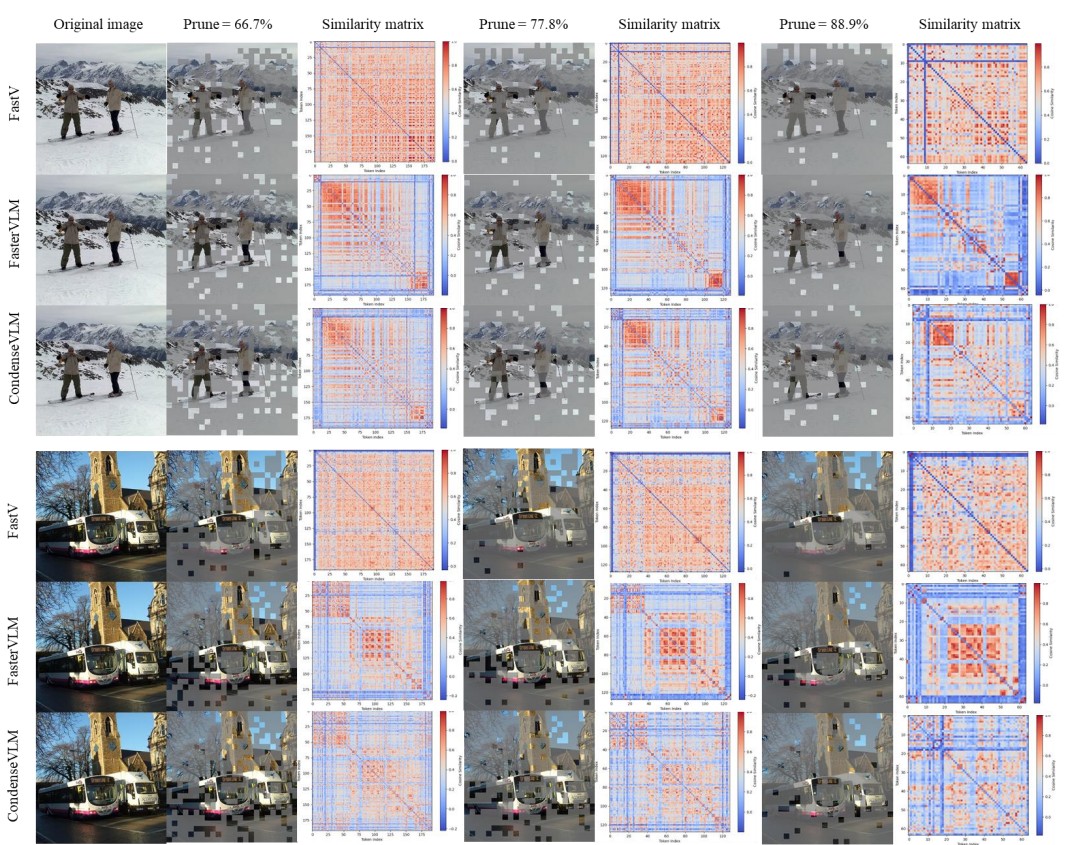

Figure 8: More case comparison between FastV, FasterVLM and CondenseVLM. It presents original images alongside their pruned versions at pruning rates of 66.7%, 77.8%, and 88.9%.

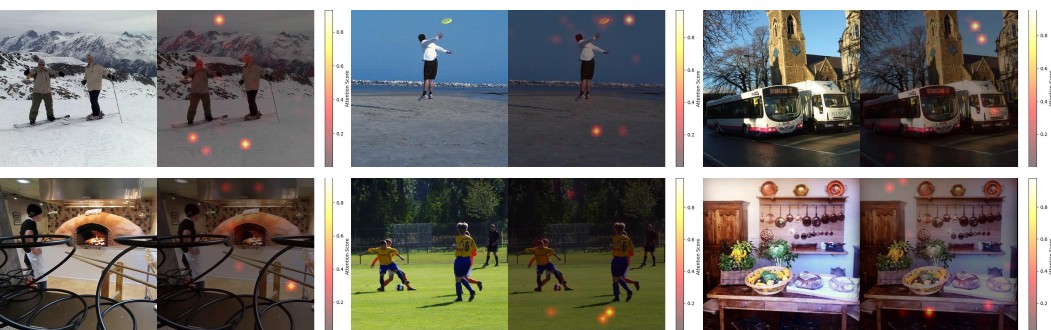

Figure 9: The distribution of [CLS] attention in different images.

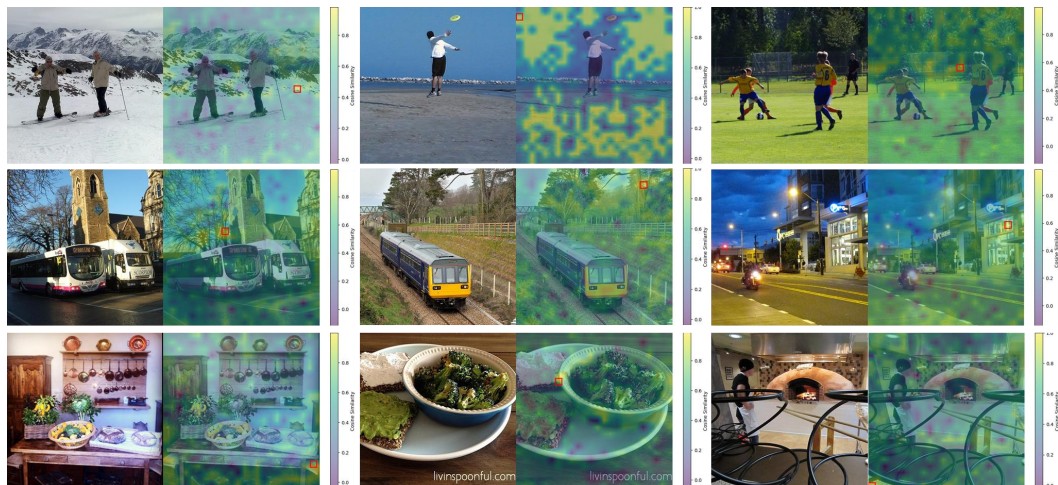

Figure 10: Visualization of similarity between background token and other tokens in different cases.

Table 15: MMBench L2-category fine-grained comparison between FasterVLM and CondenseVLM at high pruning ratios.

| Category (dev) | Vanilla | FasterVLM (↓88.9%) | Condense-VLM (↓88.9%) | FasterVLM (↓75%) | Condense-VLM (↓75%) |
|---|---|---|---|---|---|
| Attribute Reasoning | 70.4 | 70.9 | 76.4 | 75.4 | 77.8 |
| Coarse Perception | 77.4 | 73.6 | 73.2 | 77.0 | 74.9 |
| Finegrained Perception (Cross-Instance) | 55.2 | 55.2 | 54.7 | 56.6 | 55.9 |
| Finegrained Perception (Instance-Level) | 65.9 | 64.5 | 63.6 | 64.5 | 64.6 |
| Logic Reasoning | 33.1 | 30.5 | 35.8 | 30.5 | 37.0 |
| Relation Reasoning | 57.4 | 43.5 | 53.6 | 44.3 | 56.4 |

### E.2 SIMILARITY DISTRIBUTION BETWEEN BACKGROUND TOKEN AND OTHERS IN DIFFERENT SCENARIOS

Figure 10 illustrates the cosine similarity distribution between background tokens (indicated by red boxes) and other tokens across various scenes. We observe instances where tokens representing distinct semantic meanings exhibit unexpectedly high similarity scores. For example, when the selected background token represents "sky," a token representing "beach" may exhibit a higher similarity score with the sky token than tokens representing sky located in its immediate vicinity. This highlights a limitation of relying solely on similarity as a merging criterion, potentially leading to the erroneous fusion of "beach" tokens with "sky" tokens. However, visual inspection reveals that tokens representing similar elements tend to be spatially proximate. Therefore, incorporating a distance factor into the similarity metric – decreasing similarity for distant tokens and increasing similarity for proximate tokens – constitutes a more rational merging strategy

## F    MMBENCH FINEGRAINED RESULTS

As shown in Table 14, in the MMBench (Liu et al., 2025b) fine-grained comparison between FasterVLM (Zhang et al., 2024a) and CondenseVLM at 88.9% and 75% pruning ratios, significant performance improvements are evident with CondenseVLM in several categories. Specifically, CondenseVLM shows enhanced outcomes in Action Recognition, Function Reasoning, Future Prediction, Identity Reasoning, Image Quality, Image Topic, Nature Relation, OCR, Physical Property Reasoning, Physical Relation and Social Relation. These results underline CondenseVLM's ability to retain crucial visual information for complex understanding and response capabilities within dynamic environments.

The MMBench (Liu et al., 2025b) L2-category comparison in Table 15 reveals that CondenseVLM generally improves performance in categories such as Attribute Reasoning and Relation Reasoning. This indicates that CondenseVLM 's compression approach could be beneficial for detailed perceptual analysis.

## G  LIMITATION

One limitation of our work lies in its comparatively weaker performance on text-related tasks, as evidenced in Table 7. While our intention was to preserve as much information as possible by merging remaining tokens, the inclusion of critical, yet immutable, tokens in the merging process can inadvertently alter these key tokens, leading to a performance decline. Furthermore, despite our method's success in reducing redundancy among important tokens, the exhaustive search algorithm we employ for token selection introduces a noticeable computational overhead when selecting a large number of tokens. This can result in a slight reduction in inference speed. Nevertheless, even at high reduction rates, our method maintains excellent performance and relatively fast inference speeds across the vast majority of benchmarks.

## H  FUTURE WORKS

As can be observed from Table 2, in certain cases, token pruning contributes to the reduction of hallucinations. Our method achieved better results than the vanilla model on the POPE benchmark, which is specifically designed for evaluating the hallucination issues of multimodal large language models. Therefore, we believe that it is worth exploring in the future why token pruning is beneficial for reducing hallucinations and how we can better utilize efficient techniques (e.g., token pruning, and token merge) to reduce hallucinations while achieving acceleration benefits.

## I   IMPACT STATEMENT

Our work on CondenseVLM has the potential to significantly impact the field of multimodal learning and its applications. By introducing a novel token selection and merging strategy, we pave the way for more efficient and scalable multimodal models. We plan to release our code and models to the community upon acceptance, fostering further research and development in this area. The broader societal impact of our work can be seen in several key areas:

First, CondenseVLM's enhanced efficiency makes multimodal AI more accessible and deployable in resource-constrained environments, such as mobile devices and edge computing platforms. This could democratize access to powerful multimodal AI applications for users in underserved communities who may not have access to high-end computing infrastructure.

Second, by reducing the computational burden of large multimodal models, CondenseVLM promotes more sustainable AI practices. Training and deploying large-scale models consumes significant energy and resources, contributing to carbon emissions. Our work provides a pathway towards "greener" AI solutions that are more environmentally responsible.

Third, the token selection and merging strategies developed in this work could be adapted to improve the robustness and reliability of multimodal AI systems. By reducing redundancy and highlighting key features, CondenseVLM-inspired techniques could make models less susceptible to noise and adversarial attacks, leading to more trustworthy AI applications.

Finally, while our work offers promising benefits, it is crucial to acknowledge potential risks and biases. Like any AI system, CondenseVLM could be susceptible to biases present in the training data. It is therefore essential to carefully evaluate and mitigate these biases to ensure fairness and equity in its application. We are committed to responsible AI development and will actively work to address these challenges in future research.

## J   THE USE OF LARGE LANGUAGE MODELS (LLMS)

In preparing this manuscript, we utilized Gemini-2.5 pro and Doubao as a writing and editing assistant. Its role was limited to enhancing the clarity and fluency of the English in various sections. All scientific ideas, research methodology, experimental design, result analysis, and technical contributions are solely the product of the human authors. DeepSeek was not involved in any aspect of research conception, algorithm design, data interpretation, or validation of mathematical formulations, theoretical analyses, and experimental results.

