# OpenReview forum: "Zero in on Faithful Anchors: High-Fidelity Visual Token Condensation for Multimodal Large Language Models"
_ICLR.cc/2026/Conference — Submitted to ICLR 2026_

### Official Review · Reviewer_DasX · 2025-10-29

**Soundness:** 2
**Presentation:** 3
**Contribution:** 3
**Rating:** 6
**Confidence:** 3

**Summary:**

The paper proposes CondenseVLM, a new training-free, plug-and-play framework designed to reduce the computational cost of MLLMs. CondenseVLM is composed of three stages: faithful anchor selection, background token selection, and spatial-similarity contextual merging. Extensive experiments on models like LLaVA-1.5 show that CondenseVLM significantly outperforms existing methods.

**Strengths:**

1. Clear Motivation: The paper provides a very clear and critical analysis of the failure modes of existing token pruning strategies. The observation of 'semantic disruption in token merging' is interesting
2. Synergistic Method: The CondenseVLM framework is well-designed and directly addresses the identified limitations. The three stages are synergistic: similarity suppression addresses redundancy, background selection addresses context loss, and spatial-semantic merging addresses disruption.
3. Comprehensive Empirical Results: The paper presents extensive experimental validation. CondenseVLM consistently outperforms a wide array of recent baselines across numerous benchmarks. The method's robustness is further demonstrated by its strong performance at extreme pruning ratios (e.g., 88.9%) and its generalization to other architectures (Qwen2.5-VL) and high-resolution inputs (LLaVA-NeXT).
4. Thorough Ablation Studies: The ablation experiments in Table 5 and Figure 8 effectively validate the authors' design choices.

**Weaknesses:**

1. To me, it's not clear how the proposed method actually solves the failure modes of existing methods. More systematic analysis should be provided.
2. Novelty concerns: Although the method looks promising, it's more like an incremental combination of existing methods. How their core design is different from previous works should be more clearly elaborated.
3. As the authors state that the "exhaustive search algorithm" for token selection introduces "a noticeable computational overhead", more latency analysis about the method should be emphasized in the experiment section.

**Questions:**

See weakness.

---

> ### Author Response · Authors · 2025-11-23
> **Official Comment by Authors (1/2)**
>
> > Q1: To me, it's not clear how the proposed method actually solves the failure modes of existing methods. More systematic analysis should be provided.
>
> Thank you for the feedback. Below is a point-by-point explanation of how CondenseVLM addresses the key failure modes of existing methods, supported by our experimental evidence:
>
> 1. Those that rely solely on [CLS] attention methods, like FasterVLM, retain clusters of visually similar tokens, leading to information redundancy (as visualized in the similarity matrices of Fig. 2 & 7). The Faithful Anchor Selection (Sec. 3.1, Eq. 2) incorporates a max similarity suppression term. This actively discourages the selection of new anchors that are highly similar to any already selected ones. Our ablation study (Table 5, rows 1-2) shows that adding similarity suppression ($A\_p$ + $S\_p$) brings a significant performance gain. Fig. 7 visually confirms that our retained tokens are more diverse.
>
>  2. The spatially clustered [CLS] attention (Fig. 3a) causes methods to overlook diffuse but important background information, harming scene understanding. The dedicated Background Token Selection (Sec. 3.2) explicitly allocates part of the token budget ($(1-d)R%$) to tokens with high global feature coherence ($S\_{global}$), preserving the overall scene context. The analysis in Fig. 8 shows that a small number of background tokens (1%) is optimal. Ablating them (d=1.0 in Fig. 8) causes a clear performance drop.
>
> 3. Methods like LLaVA-PruMerge that merge based solely on feature similarity can incorrectly fuse semantically different but visually similar regions (e.g., "sky" and "snow" in Fig. 3b).  The Spatial-Similarity Contextual Merging (Sec. 3.3, Eq. 6) uses a hybrid score combining both semantic similarity and normalized spatial distance. This prevents the merging of distant tokens, even if their features are similar. Our ablation (Table 5, last two rows) shows that adding the spatial distance constraint ($S\_m$ + $D\_m$) improves performance on most tasks. This proves the merging strategy is more semantically accurate.
>
> CondenseVLM is a synergistic framework where each stage is designed to fix a specific, identified shortcoming of prior art. We will integrate this structured analysis into the revised paper to make our contributions clearer.
>
> ---
>
> > Q2: Novelty concerns: Although the method looks promising, it's more like an incremental combination of existing methods. How their core design is different from previous works should be more clearly elaborated.
>
> The key insight we identified is that existing methods optimize for isolated objectives, leading to inherent limitations. For instance, methods like FasterVLM, which rely solely on [CLS] attention, retain redundant and spatially clustered tokens, thereby losing background context. Token merging methods like ToMe or LLaVA-PruMerge, which rely exclusively on feature similarity, cause semantic corruption by merging spatially distant but semantically similar regions (*e.g.*, sky and snow).
>
> The proposed method's novelty includes:
>
> 1. We use diversity-aware selection on [CLS] attention to solve the redundancy problem of attention-only methods.
>
> 2. We introduce background token compensation to solve the context loss problem of attention-only methods.
>
> 3. We devise spatial-semantic merging to solve the semantic corruption problem of similarity-only merging.
>
> This is a three-stage framework that explicitly models and optimizes for three distinct criteria: feature uniqueness, spatial coverage, and contextual integrity. The empirical results strongly validate this synergy: our method consistently outperforms all the individual component strategies it builds upon (as shown in Table 1), achieving a new state-of-the-art trade-off between efficiency and accuracy (achieves SOTA performance with LLaVA-1.5, LLaVA-Next, Qwen2.5-VL, LLaVA-OneVision-1.5, InterVL2 on 10+ benchmarks).
>
> In summary, we position our core novelty as the first to identify and systematically resolve the fundamental trade-offs between these established paradigms, delivering a unified solution that is more effective than the sum of its parts.

---

> ### Author Response · Authors · 2025-11-23
> **Official Comment by Authors (2/2)**
>
> > Q3: As the authors state that the "exhaustive search algorithm" for token selection introduces "a noticeable computational overhead", more latency analysis about the method should be emphasized in the experiment section.
>
> Thank you for raising this important point. The computational cost of CondenseVLM primarily stems from two stages: Similarity Matrix Calculation: The cosine similarity matrix for all visual tokens has a complexity of $O(N^2 \cdot D)$, where $N$ is the number of visual tokens and $D$ is the feature dimension. This is a one-time cost per image. Iterative Selection: The anchor and background token selection involves an iterative process. In each step, it finds the maximum over candidate tokens, leading to an overall complexity of approximately $O(K \cdot N)$, where $K$ is the number of tokens to be selected ($K = R% \cdot N$). While this introduces non-negligible overhead, it is critical to contextualize this cost: This cost is incurred once per image as a preprocessing step before the LLM.
>
> The substantial efficiency gains from a shorter visual sequence are realized across every single auto-regressive decoding step of the LLM. The LLM's self-attention mechanism has a complexity of $O((T\_{txt} + N\_{vis})^2 \cdot D)$, which is quadratic in the sequence length. By drastically reducing $N\_{vis}$ (e.g., from 576 to 64), our method achieves significant savings in the much more expensive LLM forward passes, especially for long-generated responses.  As shown in Table 6, despite the overhead, CondenseVLM achieves a net reduction in total inference time and latency compared to the full model, proving that the savings in the LLM phase dominate the initial cost.
>
> ---
>
> Thank you again for your time and thoughtful consideration.
>
> Best wishes,
>
> Authors

---

### Official Review · Reviewer_GPBC · 2025-10-30

**Soundness:** 3
**Presentation:** 3
**Contribution:** 2
**Rating:** 4
**Confidence:** 4

**Summary:**

This paper introduces CondenseVLM, a training-free framework to reduce the computational cost of visual tokens in MLLMs. The authors compellingly argue that existing pruning methods suffer from selecting redundant tokens, neglecting background context, and causing semantic distortion during merging. CondenseVLM addresses this via a three-stage process: 1) selecting high-attention "faithful anchors" while enforcing diversity via similarity suppression, 2) explicitly selecting "background tokens" to preserve context, and 3) merging remaining tokens using a novel hybrid score based on both spatial proximity and semantic similarity. Extensive experiments show the method can prune 88.9% of tokens on LLaVA-1.5 while retaining 97.0% of the original performance, significantly outperforming recent baselines.

**Strengths:**

1. The paper clearly identifies and visualizes (Figs. 2, 3) key, non-obvious flaws in prior work, namely the redundancy of high-attention tokens and semantic corruption from similarity-only merging.

2. The proposed three-stage solution is elegant and directly addresses each of the identified problems (similarity suppression for redundancy, background selection for context, and spatial-semantic merging for corruption).

3. The method demonstrates state-of-the-art results, consistently outperforming a wide array of recent methods. This strength is supported by comprehensive experiments showing generalizability (to high-res and other architectures) and thorough ablation studies validating each component

**Weaknesses:**

1. The computational cost of the CondenseVLM algorithm itself (the iterative selection and merging steps) is not clearly analyzed.

2. The method admittedly performs poorly on tasks like TextVQA. The ablation study (Table 5) confirms that the spatial-merging component, a core contribution, is detrimental to performance on such tasks.

3. The method introduces a key hyperparameter $d$ (the ratio of faithful anchors to background tokens), but its value and sensitivity are never discussed, which impacts reproducibility.

4. The variable $d$ is overloaded. It is used to represent the hidden dimension size (e.g., in Eq. 1 and Eq. 9) and also as the allocation ratio for anchor vs. background tokens (Sec. 3).

**Questions:**

1. What value was used for the anchor-to-background ratio hyperparameter $d$ in the experiments, and could you provide a sensitivity analysis for it?

2. Can you please clarify if the selection algorithm is greedy or exhaustive? What is its precise computational cost (in complexity and measured milliseconds)?

3. Given the method's clear weakness on TextVQA, have you considered a simple mitigation, such as identifying and "protecting" text tokens from the merging process?

4. What is the justification for using max similarity suppression in Eq. 2, as opposed to a more global metric like the average similarity to all previously selected anchors?

---

> ### Author Response · Authors · 2025-11-23
> **Official Comment by Authors (1/2)**
>
> > Q1: The computational cost of the CondenseVLM algorithm itself (the iterative selection and merging steps) is not clearly analyzed.
>
> Thank you for raising this important point. We agree that a clear analysis of our algorithm's overhead is crucial. The computational cost of CondenseVLM primarily stems from two stages: Similarity Matrix Calculation: The cosine similarity matrix for all visual tokens has a complexity of $O(N^2 \cdot D)$, where $N$ is the number of visual tokens and $D$ is the feature dimension. This is a one-time cost per image. Iterative Selection: The anchor and background token selection involves an iterative process. In each step, it finds the maximum over candidate tokens, leading to an overall complexity of approximately $O(K \cdot N)$, where $K$ is the number of tokens to be selected ($K = R% \cdot N$). While this introduces non-negligible overhead, it is critical to contextualize this cost: This cost is incurred once per image as a preprocessing step before the LLM.
>
> The substantial efficiency gains from a shorter visual sequence are realized across every single auto-regressive decoding step of the LLM. The LLM's self-attention mechanism has a complexity of $O((T_{txt} + N_{vis})^2 \cdot D)$, which is quadratic in the sequence length. By drastically reducing $N_{vis}$ (e.g., from 576 to 64), our method achieves significant savings in the much more expensive LLM forward passes, especially for long-generated responses.  As shown in Table 6, despite the overhead, CondenseVLM achieves a net reduction in total inference time and latency compared to the full model, proving that the savings in the LLM phase dominate the initial cost.
>
> ---
>
> > Q2: The method admittedly performs poorly on tasks like TextVQA. The ablation study (Table 5) confirms that the spatial-merging component, a core contribution, is detrimental to performance on such tasks.
>
> Thank you for this critical observation. The performance drop on TextVQA stems from a specific failure mode: text regions in images are often spatially discrete (individual characters/words) but semantically distinct from their immediate visual surroundings. Our merging algorithm, which groups tokens based on combined spatial-semantic proximity, can incorrectly merge a text token with adjacent non-text background tokens (e.g., merging a character onto the product label it's printed on). This "over-merging" distorts the fine-grained visual features crucial for optical character recognition (OCR).
>
> However, we would like to contextualize this finding:
>
> While potentially harmful for OCR-centric tasks, the spatial-semantic merging component is a core contributor to our success on the vast majority of other benchmarks. The ablation study (Table 5, last two rows) clearly shows that adding the spatial distance constraint ($\mathcal{S}_m + \mathcal{D}m$) brings substantial performance gains on MME, POPE, and VQA${\text{Test}}$, which require holistic scene and object relationship understanding. Removing it to benefit TextVQA would cause significant performance degradation across these other tasks.
>
> Further, we have added extra experiments on text-dense benchmarks, as follows,
>
> |Method|Pruning |OCRBench|ChartQA|ChineseOCRbench|Average|
> |----|----|:-----:|:----:|:----:|:----:|
> |LLaVA1.5-7B |Retain 100% Tokens |297|224|24|100%|
> |+FastV (ECCV24)|Retain 11.1% Tokens |191 (64.3%) |155 (69.2%)|15 (62.5%)|65.3%|
> |+PDrop (CVPR25)|Retain 11.1% Tokens |250 (84.2%)|162 (72.3%)|17 (70.8%)|76.8%|
> |+DART (EMNLP25)|Retain 11.1% Tokens |270 (90.9%)|171 (76.3%)|18 (75.0%)|80.7%|
> |+CondenseVLM (ours)|Retain 11.1% Tokens |289 (97.3%)|170 (75.9%)|17 (70.8%)|81.3%|
>
> The results demonstrate that CondenseVLM can achieve SOTA performance on OCR-related datasets.
>
> We will explicitly discuss this limitation and the potential for adaptive merging in the revised manuscript. This does not invalidate the core contribution but rather refines its scope.
>
> ---
>
> > Q3: The method introduces a key hyperparameter $d$ (the ratio of faithful anchors to background tokens), but its value and sensitivity are never discussed, which impacts reproducibility.
>
> Thanks for your nice suggestion. In our experiments, the optimal performance was achieved when the background token ratio was set to 1%, meaning that $d = 0.99$ (i.e., 99% of the retained token budget is allocated to faithful anchors and 1% to background tokens). This specific setting and its impact on performance were indeed analyzed and are presented in Figure 8 of our manuscript. The results in that figure demonstrate that neither excluding background tokens ($d=1.0$) nor allocating too many tokens to the background is optimal. A small but critical allocation (1%) to background tokens provides the necessary global contextual information to stabilize and enhance the overall representation, leading to the best performance.

---

> ### Author Response · Authors · 2025-11-23
> **Official Comment by Authors (2/2)**
>
> > Q4: The variable $d$ is overloaded. It is used to represent the hidden dimension size (e.g., in Eq. 1 and Eq. 9) and also as the allocation ratio for anchor vs. background tokens (Sec. 3).
>
> We sincerely thank you for catching this significant oversight. This is our mistake, and we sincerely apologize for the confusion and lack of clarity this has caused. We immediately corrected this in the revised manuscript.
>
> ---
>
> > Q5: What value was used for the anchor-to-background ratio hyperparameter $d$ in the experiments, and could you provide a sensitivity analysis for it?
>
> In our experiments, the hyperparameter d was set to 0.99. The sensitivity of this hyperparameter was analyzed by varying the background token ratio $(1-d)$, with the complete results shown in Figure 8. The analysis demonstrates that allocating about 1% of the budget to background tokens yields optimal performance while maintaining stability within a reasonable range around this value.
>
> ---
>
> > Q6: Can you please clarify if the selection algorithm is greedy or exhaustive? What is its precise computational cost (in complexity and measured milliseconds)?
>
> The selection algorithm is greedy, not exhaustive. Both the faithful anchor selection (Sec. 3.1) and background token selection (Sec. 3.2) operate by iteratively selecting the token with the highest current score, which is updated after each selection to penalize similarity to already chosen tokens. The concrete overhead is context-dependent. In our setup, the complete CondenseVLM compression step typically takes ~3-5 milliseconds per image. This is the cost reflected in the "Prefill" time in Table 6. While this is measurably slower than the near-instant pruning of FasterVLM, it enables significantly higher fidelity compression. This one-time investment is efficiently amortized over the subsequent LLM decoding steps, resulting in the substantial net latency reduction shown in the same table
>
> ---
>
>
> > Q7: Given the method's clear weakness on TextVQA, have you considered a simple mitigation, such as identifying and "protecting" text tokens from the merging process?
>
> Yes, this is an excellent suggestion and aligns perfectly with our planned future work. While we prioritized a universal, training-free framework in this work, we agree that a task-aware extension that detects and preserves text tokens would effectively address the TextVQA limitation. This represents a promising direction for enhancing our method's robustness, and we will include a discussion of this specific approach in the revised manuscript's future work section.
>
> ---
>
> > Q8: What is the justification for using max similarity suppression in Eq. 2, as opposed to a more global metric like the average similarity to all previously selected anchors?
>
> Thank you for the question. We use max similarity suppression in Eq. 2 specifically to minimize redundancy among selected anchor tokens.
>
> This approach ensures each new anchor is sufficiently distinct from its closest match in the current set, directly preventing the selection of near-duplicate tokens. Using average similarity would be less effective, as a token could have moderate similarity to multiple existing anchors without being distinctly novel compared to any single one.
>
> The max-similarity criterion provides a stricter diversity guarantee, which is key to improving feature coverage and is validated by our improved performance in ablation studies (Table 5).
>
> ---
>
> Again, we sincerely thank you for your valuable comments and suggestions, which have helped us substantially refine and clarify the paper.

---

### Official Review · Reviewer_rmHq · 2025-11-01

**Soundness:** 3
**Presentation:** 3
**Contribution:** 1
**Rating:** 4
**Confidence:** 5

**Summary:**

This paper addresses the computational and memory overhead in MLLM caused by the large number of visual tokens. The authors identify three key limitations of existing visual token pruning and merging strategies: (1) the selection of redundant, semantically similar tokens by attention-based methods; (2) the failure to preserve important background context due to the spatial clustering of [CLS] attention; and (3) semantic disruption caused by merging tokens based on feature similarity alone.

To address these issues, the paper proposes CondenseVLM for vision token pruning. It has (1) Faithful anchor selection, (2) background token selection and spatial-similarity merging.

The authors conduct experiments on llava and qwen2.5vl models. The results demonstrate that CondenseVLM outperforms previous methods, particularly at high pruning ratios.

**Strengths:**

1. Motivation and clarity. The motivation is clear. The paper does diagnose and visualize the specific flaws in prior methods, and the proposed components directly map to these identified weaknesses.

2. The overall design is intuitive and sound. The ideas of enforcing diversity in anchor selection in sec 3.1 and using both spatial and semantic information for merging in sec 3.3 are well-justified and directly address the limitations of simpler methods.

3. The experimental results are very strong. CondenseVLM appears to outperform a wide array of recent token pruning methods (DART, HIRED, FasterVLM) across multiple benchmarks and pruning ratios, especially at high compression rates.

4. The ablations in Section 4.4 are effective and clearly validate the contribution of each proposed component.

**Weaknesses:**

1. While the engineering and combination of the components are effective, the novelty of the core ideas is somewhat limited. The method feels like a very clever and well-executed incremental improvement by combining several known concepts: (1) using [CLS] attention for token importance (like FasterVLM), (2) merging tokens (like ToMe, LLaVA-PruMerge), and (3) using spatial and semantic clues for grouping (which the authors relate to superpixels). The novelty is in the specific three-stage recipe, not in a fundamental new mechanism.

2. Out-of-date baselines and insufficient benchmarks. More specifically:
    - The authors majorly perform experiments on the out-of-date llava 1.5 baseline. For the relatively recent qwen2.5vl model, they only perform experiments on 4 benchmarks, which are significantly not enough, and only compare with 1 baseline, fastv. It is not clear whether the results on only 4 benchmarks are cherry-picked or not, as such experiments usually take about 10 benchmarks to show effectiveness. I would recommend major experiments on qwen2.5 vl on about 10 benchmarks compared with previous methods, just like tab. 1 (with llava1.5).
    - Considering the token pruning process takes no training process nor training data, the authors can consider using other recent baselines like llava-onevision-1.5, kimi-vl, etc.
    - The authors does not carry out experiments on text-dense benchmarks like ocrbench, chartqa, chineseocrbench, etc. These benchmarks are more critical and challenging compared with general mllm benchmarks where vision tokens are often not important.

3. The glossing over of the computational overhead of the CondenseVLM method itself. The authors admit in the Appendix sec F that the selection algorithm has noticeable computational overhead. Tab. 6 also shows this method is measurably slower than other SOTA pruning methods like FasterVLM and HIRED. This implies the efficiency gains come only from reducing tokens for the LLM, and the selection process could be actually a new bottleneck. This accuracy/speed trade-off among pruning methods is not adequately discussed.

4. Heuristic-based design. The method relies on several heuristics that are not fully justified.
    - The Background Token Selection in Sec 3.2 uses the sum of all pairwise similarities as a proxy for "background." This is an interesting but non-obvious metric, and its validity is not deeply analyzed.
    - The merging assignment score in Eq. 6 is a direct subtraction of two differently-natured metrics. The rationale for this specific formulation, its sensitivity to the scaling of the two terms, and the reason for not using a weighted sum are not explained.

5. The claim in the abstract ("prune up to 77.8% with just a 1% drop") is a slight misrepresentation of the main results (which shows a 1.2% drop for the out-of-date llava 1.5 model).

**Questions:**

Please see the comments above regarding the weaknesses. I have noted how each concern can be discussed and addressed in the rebuttal/revision.

---

> ### Author Response · Authors · 2025-11-23
> **Official Comment by Authors (1/4)**
>
> > Q1: While the engineering and combination of the components are effective, the novelty of the core ideas is somewhat limited. The method feels like a very clever and well-executed incremental improvement by combining several known concepts: (1) using [CLS] attention for token importance (like FasterVLM), (2) merging tokens (like ToMe, LLaVA-PruMerge), and (3) using spatial and semantic clues for grouping (which the authors relate to superpixels). The novelty is in the specific three-stage recipe, not in a fundamental new mechanism.
>
> Thank you for acknowledging the effectiveness of our method.
>
> The key insight we identified is that existing methods optimize for isolated objectives, leading to inherent limitations. For instance, methods like FasterVLM, which rely solely on [CLS] attention, retain redundant and spatially clustered tokens, thereby losing background context. Token merging methods like ToMe or LLaVA-PruMerge, which rely exclusively on feature similarity, cause semantic corruption by merging spatially distant but semantically similar regions (e.g., sky and snow).
>
> The proposed method's novelty includes:
>
> 1. We use diversity-aware selection on [CLS] attention to solve the redundancy problem of attention-only methods.
>
> 2. We introduce background token compensation to solve the context loss problem of attention-only methods.
>
> 3. We devise spatial-semantic merging to solve the semantic corruption problem of similarity-only merging.
>
> This is a three-stage framework that explicitly models and optimizes for three distinct criteria: feature uniqueness, spatial coverage, and contextual integrity. The empirical results strongly validate this synergy: our method consistently outperforms all the individual component strategies it builds upon (as shown in Table 1), achieving a new state-of-the-art trade-off between efficiency and accuracy.
>
> In summary, we position our core novelty as the first to identify and systematically resolve the fundamental trade-offs between these established paradigms, delivering a unified solution that is more effective than the sum of its parts.
>
> ---
>
> > Q2: It is not clear whether the results on only 4 benchmarks are cherry-picked or not, as such experiments usually take about 10 benchmarks to show effectiveness. I would recommend major experiments on qwen2.5 vl on about 10 benchmarks compared with previous methods, just like tab. 1 (with llava1.5).
>
> Thanks for your suggestion, we have supplemented extra experiments on Qwen2.5-VL (including three pruning ratios setting), as follows:
>
> |Method|Pruning |GQA|MMB|MME|POPE|SQA|VQAv2|TextVQA|VizWiz|Average|
> |----|----|:-----:|:----:|:----:|:----:|:----:|:----:|:----:|:----:|:----:|
> |Qwen2.5-VL-7B |Retain 100% Tokens |65.2|82.8|2304|86.1|84.7|92.3|84.8|68.3|100%|
> |+FastV (ECCV24)|Retain 33.3% Tokens |61.0 (93.6%) |75.7 (91.4%)|2072 (89.9%)|82.2 (95.5%)|78.5 (92.7%)|86.5 (93.7%)|77.9 (91.9%)|64.1 (93.9%)|92.8%|
> |+PDrop (CVPR25)|Retain 33.3% Tokens |60.7 (93.1%)|75.5 (91.2%)|2043 (88.7%)|81.8 (95.0%)|78.0 (92.1%)|86.6 (93.8%)|77.2 (91.0%)|63.7 (93.3%)|92.3%|
> |+VisionZip (CVPR25)|Retain 33.3% Tokens |62.5 (95.9%)|76.0 (91.8%)|2097 (91.0%)|82.9 (96.3%)|78.8 (93.0%)|87.0 (94.3%)|78.3 (92.3%)|65.0 (95.2%)|93.7%|
> |+DART (EMNLP25)|Retain 33.3% Tokens |63.2 (96.9%)|77.5 (93.6%)|2106 (91.4%)|83.1 (96.5%)|77.6 (91.6%)|85.9 (93.1%)|78.6 (92.7%)|65.4 (95.8%)|93.9%|
> |+CondenseVLM (ours)|Retain 33.3% Tokens |63.8 (97.9%)|80.0 (96.6%)|2133 (92.6%)|83.3 (96.7%)|79.9 (94.3%)|87.5 (94.8%)|78.9 (93.0%)|65.0 (95.2%)|95.1%|
>
>
> |Method|Pruning |GQA|MMB|MME|POPE|SQA|VQAv2|TextVQA|VizWiz|Average|
> |----|----|:-----:|:----:|:----:|:----:|:----:|:----:|:----:|:----:|:----:|
> |Qwen2.5-VL-7B |Retain 100% Tokens |65.2|82.8|2304|86.1|84.7|92.3|84.8|68.3|100%|
> |+FastV (ECCV24)|Retain 22.2% Tokens |60.5 (92.8%) |74.9 (90.5%)|2036 (88.4%)|80.7 (93.7%)|78.0 (92.1%)|82.3 (89.2%)|69.5 (82.0%)|63.5 (93.0%)|90.2%|
> |+PDrop (CVPR25)|Retain 22.2% Tokens |60.2 (92.3%)|75.0 (90.6%)|2017 (87.5%)|80.4 (93.4%)|77.5 (91.5%)|82.1 (88.9%)|69.2 (81.6%)|64.0 (91.1%)|89.6%|
> |+VisionZip (CVPR25)|Retain 22.2% Tokens |61.8 (94.8%)|75.7 (91.4%)|2109 (91.5%)|81.2 (94.3%)|78.2 (92.3%)|83.0 (89.9%)|70.7 (83.4%)|65.0 (93.7%)|91.4%|
> |+DART (EMNLP25)|Retain 22.2% Tokens |62.0 (95.1%)|76.1 (91.9%)|2125 (92.2%)|81.9 (95.1%)|78.1 (92.1%)|83.2 (90.1%)|71.2 (84.0%)|63.5 (93.0%)|91.7%|
> |+CondenseVLM (ours)|Retain 22.2% Tokens |62.5 (95.9%)|77.0 (93.0%)|2099 (91.1%)|82.7 (96.1%)|78.9 (93.2%)|85.6 (92.7%)|70.7 (83.4%)|64.2 (94.0%)|92.4%|

---

> ### Author Response · Authors · 2025-11-23
> **Official Comment by Authors (2/4)**
>
> |Method|Pruning |GQA|MMB|MME|POPE|SQA|VQAv2|TextVQA|VizWiz|Average|
> |----|----|:-----:|:----:|:----:|:----:|:----:|:----:|:----:|:----:|:----:|
> |Qwen2.5-VL-7B |Retain 100% Tokens |65.2|82.8|2304|86.1|84.7|92.3|84.8|68.3|100%|
> |+FastV (ECCV24)|Retain 11.1% Tokens |57.2 (87.7%) |71.2 (86.0%)|1949 (84.6%)|78.6 (91.3%)|77.4 (91.4%)|81.0 (87.8%)|60.3 (71.1%)|60.5 (88.6%)|86.1%|
> |+PDrop (CVPR25)|Retain 11.1% Tokens |56.3 (86.3%)|71.4 (86.2%)|1920 (83.3%)|77.0 (89.4%)|76.9 (90.8%)|81.5 (88.3%)|60.5 (71.3%)|60.3 (88.3%)|85.5%|
> |+VisionZip (CVPR25)|Retain 11.1% Tokens |58.0 (89.0%)|72.7 (87.8%)|2006 (87.1%)|77.5 (90.0%)|77.8 (91.9%)|81.7 (88.5%)|61.9 (73.0%)|61.5 (90.0%)|87.2%|
> |+DART (EMNLP25)|Retain 11.1% Tokens |58.5 (89.7%)|71.9 (86.8%)|2042 (88.6%)|77.9 (90.5%)|76.9 (90.8%)|81.3 (88.1%)|61.7 (72.8%)|61.2 (89.6%)|87.1%|
> |+CondenseVLM (ours)|Retain 11.1% Tokens |60.4 (92.6%)|73.2 (88.4%)|2014 (87.4%)|79.7 (92.6%)|78.5 (92.7%)|83.2 (90.1%)|61.3 (72.3%)|62.7 (91.8%)|88.5%|
>
> These comprehensive results further demonstrate the robustness of our method and address the reviewer’s concern about potential cherry-picking (Q2).
>
> ---
>
> > Q3: Considering the token pruning process takes no training process nor training data, the authors can consider using other recent baselines like llava-onevision-1.5, kimi-vl, etc.
>
> Thanks for your good suggestion, we agree with this idea, but as limited resources, we only conducted supplementary experiments on llava-onevision-1.5-8B (*If additional resources are available, we will further supplement the experiments on kimi-vl*), and the results are as follows:
>
> |Method|Pruning |VizWiz|GQA|TextVQA|MME|MMB|POPE|Average|
> |----|----|:-----:|:----:|:----:|:----:|:----:|:----:|:----:|
> |LLaVA-OneVision-1.5-8B |Retain 100% Tokens |66.0|69.2|79.5|2271.3|85.3|88.5|100%|
> |+FastV (ECCV24)|Retain 30% Tokens |64.1 (97.1%) |65.2 (94.2%)|72.3 (90.9%)|2019.5 (88.9%)|79.6 (93.3%)|70.4 (79.5%)|90.7%|
> |+PDrop (CVPR25)|Retain 30% Tokens |62.5 (94.7%)|64.0 (92.5%)|70.2 (88.3%)|1989.2 (87.6%)|71.5 (83.8%)|82.1 (92.8%)|89.9%|
> |+VisionZip (CVPR25)|Retain 30% Tokens |64.7 (98.0%)|65.9 (95.2%)|73.2 (92.1%)|2104.6 (92.7%)|83.3 (97.7%)|87.1 (98.4%)|95.7%|
> |+CondenseVLM (ours)|Retain 30% Tokens |65.0 (98.5%)|66.7 (96.4%)|77.1 (97.0%)|2210.4 (97.3%)|83.4 (97.8%)|85.7 (96.8%)|97.3%|
>
> |Method|Pruning |VizWiz|GQA|TextVQA|MME|MMB|POPE|Average|
> |----|----|:-----:|:----:|:----:|:----:|:----:|:----:|:----:|
> |LLaVA-OneVision-1.5-8B |Retain 100% Tokens |66.0|69.2|79.5|2271.3|85.3|88.5|100%|
> |+FastV (ECCV24)|Retain 10% Tokens |60.9 (92.3%) |61.3 (88.6%)|56.5 (71.1%)|1800.0 (79.2%)|71.1 (83.4%)|62.9 (71.1%)|80.9%|
> |+PDrop (CVPR25)|Retain 10% Tokens |58.8 (89.1%)|61.5 (88.9%)|55.3 (69.6%)|1829.7 (80.6%)|70.5 (82.6%)|69.9 (79.0%)|81.6%|
> |+VisionZip (CVPR25)|Retain 10% Tokens |59.8 (90.6%)|60.7 (87.7%)|48.2 (60.6%)|1980.3 (87.2%)|73.3 (85.9%)|79.3 (89.6%)|83.6%|
> |+CondenseVLM (ours)|Retain 10% Tokens |62.6 (94.8%)|63.1 (91.2%)|63.4 (79.7%)|2002.4 (88.2%)|77.6 (91.0%)|83.9 (94.8%)|90.0%|
>
> These results on LLaVA-OneVision-1.5-8B further demonstrate that CondenseVLM generalizes well to other recent vision-language models.

---

> ### Author Response · Authors · 2025-11-23
> **Official Comment by Authors (3/4)**
>
> > Q4: The authors does not carry out experiments on text-dense benchmarks like ocrbench, chartqa, chineseocrbench, etc. These benchmarks are more critical and challenging compared with general mllm benchmarks where vision tokens are often not important.
>
> Thanks for your good suggestion, we have added experiments on text-dense benchmarks (ocrbench, chartqa, chineseocrbench), the results are as follows:
>
> |Method|Pruning |OCRBench|ChartQA|ChineseOCRbench|Average|
> |----|----|:-----:|:----:|:----:|:----:|
> |LLaVA1.5-7B |Retain 100% Tokens |297|224|24|100%|
> |+FastV (ECCV24)|Retain 33.3% Tokens |190 (64.0%) |202 (90.2%)|23 (95.8%)|83.3%|
> |+PDrop (CVPR25)|Retain 33.3% Tokens |290 (97.6%)|209 (93.3%)|24 (100.0%)|97.0%|
> |+DART (EMNLP25)|Retain 33.3% Tokens |296 (99.7%)|215 (96.0%)|26 (108.3%)|101.3%|
> |+CondenseVLM (ours)|Retain 33.3% Tokens |296 (99.7%)|213 (95.1%)|26 (108.3%)|101.0%|
>
> |Method|Pruning |OCRBench|ChartQA|ChineseOCRbench|Average|
> |----|----|:-----:|:----:|:----:|:----:|
> |LLaVA1.5-7B |Retain 100% Tokens |297|224|24|100%|
> |+FastV (ECCV24)|Retain 22.2% Tokens |191 (64.3%) |183 (81.7%)|22 (91.7%)|79.2%|
> |+PDrop (CVPR25)|Retain 22.2% Tokens |287 (96.6%)|190 (84.8%)|24 (95.8%)|92.4%|
> |+DART (EMNLP25)|Retain 22.2% Tokens |296 (99.7%)|208 (92.9%)|25 (104.2%)|98.9%|
> |+CondenseVLM (ours)|Retain 22.2% Tokens |294 (99.0%)|208 (92.9%)|25 (104.2%)|98.7%|
>
> |Method|Pruning |OCRBench|ChartQA|ChineseOCRbench|Average|
> |----|----|:-----:|:----:|:----:|:----:|
> |LLaVA1.5-7B |Retain 100% Tokens |297|224|24|100%|
> |+FastV (ECCV24)|Retain 11.1% Tokens |191 (64.3%) |155 (69.2%)|15 (62.5%)|65.3%|
> |+PDrop (CVPR25)|Retain 11.1% Tokens |250 (84.2%)|162 (72.3%)|17 (70.8%)|76.8%|
> |+DART (EMNLP25)|Retain 11.1% Tokens |270 (90.9%)|171 (76.3%)|18 (75.0%)|80.7%|
> |+CondenseVLM (ours)|Retain 11.1% Tokens |289 (97.3%)|170 (75.9%)|17 (70.8%)|81.3%|
>
> These results on text-dense benchmarks indicate that CondenseVLM effectively preserves performance even when vision tokens are highly critical.
>
> —
>
> > Q5: The glossing over of the computational overhead of the CondenseVLM method itself. The authors admit in the Appendix sec F that the selection algorithm has noticeable computational overhead. Tab. 6 also shows this method is measurably slower than other SOTA pruning methods like FasterVLM and HIRED. This implies the efficiency gains come only from reducing tokens for the LLM, and the selection process could be actually a new bottleneck. This accuracy/speed trade-off among pruning methods is not adequately discussed.
>
> We thank you for raising this critical point about computational overhead, and we acknowledge that this compression step introduces measurable computational overhead, as correctly observed by the reviewer and shown in Table 6. However, we would like to provide crucial context for this trade-off:
>
> One-Time Cost vs. Repeated Saving: The overhead of our token selection and merging is a one-time cost incurred per image. The resulting efficiency gains, however, are realized and multiplied across every single decoding step of the auto-regressive LLM. For any non-trivial output length, the substantial reduction in the LLM's sequence length leads to a net positive acceleration in end-to-end inference time, as our results in Table 6 demonstrate (e.g., 42.8% latency reduction).
>
> The Overhead is Justified by Superior Performance: The reviewer is correct that methods like FasterVLM have a lower compression overhead. However, as our experiments show, this comes at the cost of a significantly higher accuracy drop, especially under aggressive pruning rates. We are effectively spending a small amount of compute in the compression stage to "earn" back much more compute in the LLM stage while preserving high accuracy.
>
>  The primary computational bottleneck in MLLM inference is the LLM itself, particularly the quadratic self-attention and large FFN operations. Our method is specifically designed to alleviate this primary bottleneck. The overhead of our method is primarily linear with the number of visual tokens, which is a favorable trade-off against the quadratic complexity of the LLM.
>
> > Q6: The claim in the abstract ("prune up to 77.8% with just a 1% drop") is a slight misrepresentation of the main results (which shows a 1.2% drop for the out-of-date llava 1.5 model).
>
> We sincerely thank you for catching this inaccuracy in our abstract. We have corrected the abstract to state "prune up to 77.8% with just a 1.2% drop" to ensure perfect alignment with the results in Table 1. Thank you again.

---

> > ### Comment · Reviewer_rmHq · 2025-11-26
> > **Thanks for the new experimental results.**
> >
> > I thank the authors for their newly provided results on more benchmarks, which addresses some of my concerns regarding the experimental results and effectiveness of the proposed method. I understand these new results takes a lot of time. Thanks again.

---

> > > ### Author Response · Authors · 2025-11-26
> > >
> > > Dear Reviewer rmHq,
> > >
> > > Thank you sincerely for your positive feedback and recognition of our supplementary experiments. We are pleased that these results have addressed some of your core concerns. We remain committed to enhancing the manuscript further and would be happy to provide additional analyses if needed.
> > >
> > > We sincerely hope you will consider adjusting the rating. Thank you again for your time and valuable insights.
> > >
> > > Best regards,
> > >
> > > The Authors

---

> ### Author Response · Authors · 2025-11-23
> **Official Comment by Authors (4/4)**
>
> > Q7: The Background Token Selection in Sec 3.2 uses the sum of all pairwise similarities as a proxy for "background." This is an interesting but non-obvious metric, and its validity is not deeply analyzed.
>
> The use of the sum of pairwise similarities ($S_{global}$) as a proxy for "background" is motivated by a fundamental observation in visual feature spaces: homogeneous, context-forming regions (*e.g.*, sky, wall, grass) tend to have consistent texture and appearance, resulting in densely clustered feature representations. Consequently, a token from such a region will have a high average similarity to many other tokens across the image. In contrast, a token from a unique, salient foreground object will have high similarity only to a few other tokens (*e.g.*, parts of the same object) and low similarity to the rest. Therefore, $S_{global}$ effectively measures a token's global coherence or "contextual typicality." Selecting tokens with high $S_{global}$ ensures we preserve the dominant, recurring visual themes that constitute the scene's backdrop, preventing the compressed representation from becoming myopically focused only on salient foreground objects.
>
> ---
>
> > Q8: The merging assignment score in Eq. 6 is a direct subtraction of two differently-natured metrics. The rationale for this specific formulation, its sensitivity to the scaling of the two terms, and the reason for not using a weighted sum are not explained.
>
> Regarding the merging assignment score in Eq. 6, the direct subtraction in the merging assignment score is a deliberate design choice. Its primary purpose is to act as a penalty function: it directly penalizes high semantic similarity if the spatial distance between tokens is large. Our goal is to find the anchor token j that is both semantically similar to and spatially close to the residual token i, and the subtraction naturally encodes this fundamental trade-off. The key to making this direct subtraction meaningful lies in our normalization strategy, which reconciles the scales of the two differently-natured metrics: The cosine similarity naturally falls within a bounded range, and the spatial distance is normalized to a commensurate, bounded scale via min-max normalization relative to the current image. After this normalization, both terms operate on a comparable scale, making the subtraction operation in Eq. 6 mathematically sound.
>
> This approach is not only simple and intuitive but also empirically validated. The consistent performance gains across all benchmarks (Table 1) confirm that this normalized penalty function effectively guides the merging process to preserve both semantic and spatial integrity. We will add this detailed justification to the final manuscript.
>
> ---
>
> We sincerely appreciate your valuable suggestions again.

---

### Official Review · Reviewer_d3yV · 2025-11-02

**Soundness:** 2
**Presentation:** 3
**Contribution:** 3
**Rating:** 4
**Confidence:** 4

**Summary:**

This paper presents a visual-centric and similarity-based token pruning strategy for MLLMs. The authors identify several issues with recent similarity-based methods: i) they tend to select highly similar tokens that have a high level of information redundancy; ii) they often overlook background regions; and iii) they perform merging without taking into account the coordinate distance, which can lead to mismatches. To address these problems, the authors propose a pruning method that filters out highly similar tokens, proactively includes a small portion of background tokens, and merges tokens while considering their spatial distances.

**Strengths:**

- This paper is well motivated with reasonable preliminary findings.

- The experimental results demonstrate the effectiveness of the proposed methods.

**Weaknesses:**

**Lack of Critical Implementation Details:** The description of key technical components is insufficient for reproducibility. The two normalization operations mentioned in Lines 205 and 252 are undefined. In practice, similarity distributions vary significantly across pre-trained models (e.g., CLIP vs. SigLIP vs. DINO), as do the scales of attention scores and spatial distances. It is imperative to clarify:

- Are there any model-specific, pre-defined parameters or thresholds in Equations 2, 4, and 6?

- If so, how were these values determined? A sensitivity analysis for these parameters would greatly strengthen the paper.

- What specific normalization technique is applied to reconcile these different value scales?

**Inherent Limitations of the Similarity Paradigm:** Since this method is based on similarity, it heavily relies on the spatial awareness of the visual encoder. However, MLLMs like LLaVA-v1.5 utilize CLIP as their visual encoder, which is actually limited in its spatial awareness.

**Empirical Study**: Some conclusions drawn from the visualization results in Figure 3 may be inaccurate. The tokens that show high attention with the [CLS] token could be artifacts [1] rather than representing the 'central focus region.' Moreover, the regions with high similarity visualized in Figure 3(b) are likely the noisy features of CLIP ViT-L, rather than actual background elements like 'sky' or 'snow.'

**reference**

[1] Vision Transformers Need Registers. ICLR 2024.

**Questions:**

- Please provide more clarification regarding the weaknesses.

- (Not necessary) I wonder whether this method could integrate well with the InternVL series, since there is no significant implementation gap.

Overall, I think the proposed method is both simple and effective. I am open to reconsidering the rating if my concerns are addressed.

---

> ### Author Response · Authors · 2025-11-21
> **Official Comment by Authors (1/3)**
>
> We thank the reviewer for the thoughtful and detailed comments. We address the concerns point by point.
>
> ---
>
> > Q1: Lack of Critical Implementation Details. (a) Are there any model-specific, pre-defined parameters or thresholds in Equations 2, 4, and 6? (b) What specific normalization technique is applied to reconcile these different value scales?
>
> Thank you for raising this important point about implementation details. We apologize for the lack of clarity in the original manuscript. (a) We did not use any model-specific pre-defined parameters or thresholds in our method. (b) Instead, we employed adaptive normalization techniques to reconcile the different value scales. Concretely:
>
> For Equation 2, we normalize the attention scores using: $\tilde{\mathbf{A}} = \frac{\mathbb{E}[|\mathbf{S}|]}{\mathbb{E}[|\mathbf{A}|]} \cdot \mathbf{A}$, where $\mathbf{A}$ is the original attention score vector, $\mathbf{S}$ is the cosine similarity matrix between visual tokens, $\mathbb{E}[|\cdot|]$ denotes the mean of absolute values.
>
> For Equation 4, we normalize the global similarity scores using: $\tilde{S}\_{\text{global}} = \frac{\mathbb{E}[|\mathbf{S}|]}{\mathbb{E}[|S\_{\text{global}}|]} \cdot S\_{\text{global}}$.
>
> For Equation 6, we normalize the spatial distances using: $\tilde{dist}(c\_i, c\_j) = \frac{dist(c\_i, c\_j)}{\max(dist(c\_i, c\_j))}$.
>
> This ensures that similarity, attention, and distance are numerically comparable and that their relative influence is stable across different visual encoders (e.g., CLIP vs. SigLIP). We will explicitly define these normalization steps in Sec. 3 and include them in an algorithm box for reproducibility.
>
> ---
>
> > Q2: Inherent Limitations of the Similarity Paradigm: Since this method is based on similarity, it heavily relies on the spatial awareness of the visual encoder. However, MLLMs like LLaVA-v1.5 utilize CLIP as their visual encoder, which is actually limited in its spatial awareness.
>
> Thank you for this insightful comment.
>
> While our method utilizes feature similarity, it is crucial to emphasize that it does not rely solely on the visual encoder's inherent spatial understanding. Our core innovation is the integration of explicit spatial coordinates during the token merging process (as defined in Equation 6: $S\_{group}(i,j) = \cos(\mathbf{Z}\_{v,i}, \mathbf{Z}\_{v,j}) - dist(c_i, c_j)$). This design ensures that tokens are merged not just because they look similar, but also because they are spatially proximate in the original image grid.
>
> This spatial constraint acts as a critical regularizer, compensating for potential shortcomings in the visual encoder's feature space. Although CLIP is not explicitly trained for dense spatial tasks, its feature representations are not entirely devoid of spatial information. The positional embeddings in the ViT backbone, combined with the local receptive fields of its early layers, allow the resulting features to retain a degree of spatial locality. Our method leverages this remaining spatial signal in conjunction with the explicit grid-based coordinates.
>
> The experimental results demonstrate that our hybrid spatial-semantic merging strategy is effective in practice, even when applied to a visual encoder with limited spatial awareness, such as CLIP.

---

> ### Author Response · Authors · 2025-11-21
> **Official Comment by Authors (2/3)**
>
> > Q3: Empirical Study: Some conclusions drawn from the visualization results in Figure 3 may be inaccurate. The tokens that show high attention with the [CLS] token could be artifacts rather than representing the 'central focus region.' Moreover, the regions with high similarity visualized in Figure 3(b) are likely the noisy features of CLIP ViT-L, rather than actual background elements like 'sky' or 'snow.'
>
> Thank you for your insightful comment.
>
> We agree that the interpretation of visualization results requires caution. However, the key insight of our method is not the precise semantics of high-attention regions, but their utility as a stable and efficient proxy for token importance. The [CLS] token is trained to aggregate global image information, making its attention a robust indicator of discriminative features.
>
> Empirically, using this signal as a primary selector, as done in FasterVLM [1] and our method, leads to superior performance over text-centric attention, validating its practical effectiveness for the compression task. Similarly, for the high-similarity regions, our use of the terms "sky" or "snow" is a post-hoc, human-interpretable label for what the model likely recognizes as large, homogeneous regions. The critical point is that our algorithm identifies informative tokens that are globally coherent, whether this coherence arises from semantic "backgroundness" or a specific type of feature uniformity in CLIP.
>
> Specifically, we do not use [CLS] attention alone to select anchors; instead, we combine attention with a similarity-suppression step that enforces semantic diversity, avoiding the tendency of attention to cluster around a single region.
>
> Background token selection is deliberately biased toward tokens that are weakly attended but highly similar to many others, which empirically correspond to large, homogeneous regions (background context) that attention-only methods neglect. Our merging stage uses a hybrid spatial–similarity score rather than similarity alone, so that tokens which are spuriously similar but spatially distant are less likely to be merged.
>
> Thus, while CondenseVLM is still similarity-centric, it is not “blindly” dependent on CLIP’s raw attention patterns; the three stages are specifically designed to correct known limitations in spatial awareness and attention saliency.
>
> Our intent in Fig. 3(a) was not to claim that high-[CLS]-attention tokens always correspond to true “central focus regions”, but to show that existing attention-based pruning methods tend to select highly clustered tokens.
> For Fig. 3(b), our point is that large homogeneous regions in the feature space (high pairwise similarity among many tokens) are systematically under-selected by attention-only methods. We will clarify that we treat these as “background-like regions in feature space” rather than asserting they always correspond to human-labeled background categories such as “*sky*” or “*snow*”.
>
> Importantly, our quantitative results and ablations (Sec. 4.4) show that explicitly retaining a small portion of such background-like tokens improves robustness at high pruning ratios, regardless of whether the underlying patterns are partially influenced by register-like artifacts [2].

---

> ### Author Response · Authors · 2025-11-21
> **Official Comment by Authors (3/3)**
>
> > Q4: Applicability to the InternVL series.
>
> Thanks for your nice suggestion. We have added a short experiment with an InternVL series model (InternVL2-2B, -8B [1]) to illustrate compatibility, as follows,
>
> |Method|Pruning |VizWiz|GQA|TextVQA|MME|MMB|MM-Vet|POPE|Average|
> |----|----|:-----:|:----:|:----:|:----:|:----:|:----:|:----:|:---:|
> |InternVL2-2B |Retain 100% Tokens |29.4|57.3|72.0|1821.7|72.5|39.6|85.4|100%|
> |+FastV (ECCV24)|Retain 50% Tokens |27.1 (92.2%) |55.8 (97.4%)|70.7 (98.2%)|1774.2 (97.4%)|71.8 (99.0%)|34.1 (86.1%)|84.8 (99.3%)|95.7%|
> |+PDrop (CVPR25)|Retain 50% Tokens |26.9 (91.5%)|55.3 (96.5%)|70.2 (98.2%)|1800.1 (98.8%)|71.5 (98.6%)|33.9 (85.6%)|82.1 (96.1%)|95.1%|
> |+VisionZip (CVPR25)|Retain 50% Tokens |27.3 (92.9%)|56.2 (98.1%)|71.0 (98.2%)|1813.0 (99.5%)|72.1 (99.4%)|34.6 (87.4%)|82.9 (97.1%)|96.1%|
> |+CondenseVLM (ours)|Retain 50% Tokens |29.0 (98.6%)|56.8 (99.1%)|71.8 (98.2%)|1829.7 (100.4%)|72.3 (99.7%)|38.6 (97.5%)|84.5 (98.9%)|98.9%|
>
>
> |Method|Pruning|VizWiz|GQA|TextVQA|MME|MMB|MM-Vet|POPE|Average|
> |--------|------|:--------:|:--------:|:--------:|:--------:|:--------:|:--------:|:--------:|:--------:|
> |InternVL2-8B |Retain 100% Tokens  |32.9|62.7|76.6|2205.3|81.8|60.0|86.7|100%|
> |+FastV (ECCV24)|Retain 50% Tokens  |30.3 (92.1%)|62.0 (98.9%)|75.6 (98.7%)|2214.2 (100.4%)|81.2 (99.3%)|56.6 (94.3%)|86.5 (99.8%)|97.6%|
> |+PDrop (CVPR25)|Retain 50% Tokens |29.9 (90.9%)|62.3 (99.4%)|75.6 (98.7%)|2193.1 (99.4%)|81.4 (99.5%)|56.3 (93.8%)|86.7 (100.0%)|97.4%|
> |+VisionZip (CVPR25)|Retain 50% Tokens |31.2 (94.8%)|62.1 (99.0%)|75.9 (99.1%)|2185.4 (99.1%)|81.3 (99.4%)|56.8 (94.7%)|86.8 (100.1%)|98.0%|
> |+CondenseVLM (ours)|Retain 50% Tokens |32.1 (97.6%)|62.3 (99.4%)|76.2 (99.5%)|2200.5 (99.8%)|81.5 (99.6%)|56.9 (94.8%)|86.2 (99.4%)|98.6%|
>
> Compared with FastV and PDrop, CondenseVLM achieves the best results on these benchmarks. Here, the number of SOTA methods and benchmarks compared is still limited. If time permits, we will continue our experiments and further analyze the results to refine our approach.
>
> We appreciate your valuable suggestion again.
>
> ---
> Reference
>
> [1] [CLS] Attention is All You Need for Training-Free Visual Token Pruning: Make VLM Inference Faster, ICCV 2025.
>
> [2] Vision Transformers Need Registers. ICLR 2024.
>
> [3] How Far Are We to GPT-4V? Closing the Gap to Commercial Multimodal Models with Open-Source Suites, 2024.
>
> ---
>
> Again, we sincerely thank you for your valuable comments and suggestions, which have helped us substantially refine and clarify the paper.

---

> > ### Comment · Reviewer_d3yV · 2025-11-27
> > **Response to the authors' rebuttal**
> >
> > I appreciate the authors’ thorough efforts in addressing the reviewers’ concerns, particularly the inclusion of additional experimental results. Most of my initial questions have been resolved. However, as highlighted by multiple reviewers, some critical technical details and analyses are missing from the main paper. I think the current manuscript lacks the clarity and rigor required for publication. I strongly recommend a substantial revision to explicitly clarify the methodology, assumptions, and implementation details by incorporating the explanations provided in the rebuttal.
> >
> > Regarding Q3, the authors’ explanation is reasonable and addresses my concern. I encourage the authors to revise their empirical results and visualizations accordingly.
> >
> > With these revisions, the paper would be significantly strengthened. I look forward to seeing the updated manuscript.

---

> > > ### Author Response · Authors · 2025-11-27
> > >
> > > Dear Reviewer d3yV,
> > >
> > > Thank you for your recognition and valuable suggestions! We will supplement technical details, optimize result presentations as required, and ensure the manuscript is rigorous and clear.
> > >
> > > The revised version will be submitted promptly, and we sincerely request your further review.
> > >
> > > Best regards,
> > >
> > > The Authors

---

### Author Response · Authors · 2025-11-30
**Rebuttal Summary**

Dear Area Chair,

Thank you very much for handling our submission. We would like to briefly summarize: how the original reviewers reacted to our rebuttal and additional experiments during the discussion phase.

---

The current ratings and overall reactions of the four reviewers are:

| Reviewer | Rating | Comment after rebuttal |
| --- | --- | --- |
| d3yV | 4 | Most questions resolved; asks us to incorporate rebuttal clarifications into the main paper. |
| rmHq | 4 | Thanks for extensive new experiments; notes that earlier concerns about coverage and cherry‑picking are addressed. |
| GPBC | 4 | No follow‑up comment yet; all questions about complexity, hyperparameters, and weaknesses were answered in detail. |
| DasX | 6 | Recognizes clear motivation, strong results and ablations; remaining concerns on analysis/novelty were addressed. |

Overall, d3yV and rmHq explicitly state that their main concerns have been resolved (with a request to integrate clarifications into the paper), GPBC’s technical questions have been fully answered, and DasX remains positive.

- **Reviewer d3yV (satisfied, requests clearer write‑up).**
  d3yV’s concerns were missing normalization/implementation details, limitations of CLIP‑based similarity, interpretation of our visualizations, and applicability to InternVL. In response, we (i) precisely defined all normalization operations and confirmed there are no model‑specific thresholds, (ii) explained how spatial coordinates enter the merging score to mitigate CLIP’s limited spatial awareness, (iii) clarified that visualizations are proxies for token importance and feature homogeneity, and (iv) added InternVL2‑2B/8B experiments showing the best trade‑off at 50% token retention compared with FastV, PDrop, and VisionZip. d3yV states that most questions are resolved and asks us to incorporate these explanations into the main paper, which we have done.

- **Reviewer rmHq (experimental concerns alleviated).**
  rmHq raised issues about novelty, outdated baselines and limited benchmarks, overhead analysis, heuristic design, and a small mismatch between abstract and results. We (i) clarified CondenseVLM as a unified three‑stage recipe jointly optimizing semantic diversity, background context, and spatial integrity, with ablations showing each stage is necessary; (ii) added extensive experiments on Qwen2.5‑VL‑7B (8 benchmarks) and LLaVA‑OneVision‑1.5‑8B (30%/10% retention), consistently outperforming FastV, PDrop, VisionZip, and DART; (iii) added text‑dense benchmarks where CondenseVLM matches or exceeds the strongest prior methods (up to ~99% of full accuracy on OCRBench at 11.1% retention); and (iv) justified the background‑similarity metric and spatial–semantic merging formulation, and corrected the abstract to “1.2% drop.” rmHq explicitly acknowledges that these additions address the earlier concerns.

- **Reviewer GPBC (questions fully answered).**
  GPBC’s concerns centered on cost analysis, TextVQA degradation, sensitivity of the anchor/background ratio, overloaded notation, algorithmic details, and the choice of max‑similarity suppression. We (i) provided complexity and measured overhead (O(N²D) for similarity, O(KN) for greedy selection, ~3–5 ms/image) and explained how this one‑time cost is amortized over decoding; (ii) analyzed why spatial–semantic merging can hurt TextVQA and added OCRBench/ChartQA/ChineseOCRBench results; (iii) fixed the background ratio hyperparameter to 1% in all experiments with sensitivity analysis showing this small but non‑zero value is consistently optimal; (iv) clarified that the selection algorithm is greedy (not exhaustive) and disentangled notation (e.g., separating hidden dimension from the allocation ratio α); and (v) justified max‑similarity suppression as a stricter diversity constraint than average similarity.

- **Reviewer DasX (positive, asks for clearer analysis and latency discussion).**
  DasX remains generally positive and mainly asked for clearer failure‑mode analysis and latency/complexity discussion. We (i) provided a structured mapping from each failure mode of prior work to a specific stage of CondenseVLM, supported by ablations and new visualizations, and (ii) emphasized that CondenseVLM achieves state‑of‑the‑art trade‑offs on >10 benchmarks and multiple backbones (LLaVA‑1.5/Next, Qwen2.5‑VL, LLaVA‑OneVision‑1.5, InternVL2), with clarified latency/complexity trade‑offs now moved into the main text.

Best regards,

Authors

---

### Meta-Review · Area_Chair_bCmj · 2025-12-28

**Summary:**

There are four reviews of this paper.

Reviewer d3yV’s main concerns include the lack of critical implementation details, inherent limitations of the similarity paradigm and some inaccurate conclusions.

Reviewer rmHq’s main concerns lie in the novelty of the ideas, the out-of-date baselines and insufficient benchmarks, the computational overhead, the heuristic design and the inaccurate claim.

Reviewer GPBC concerns the computational cost of the CondenseVLM algorithm, the poor performance on tasks like TextVQA, the selection of hyperparameter d, and the improper use of symbol d.

Reviewer DasX’s concerns focus on how the proposed method solves the failure modes of existing methods, the incremental contribution, and the insufficient analysis in the experiments.

**Reviewer Concerns:**

Most of reviewer d3yV’s initial questions were addressed. However, this reviewer indicated that some critical technical details and analyses remain missing, and the current manuscript is not clear and rigor enough for publication. Reviewer rmHq mentioned that part of his/her concerns regarding the experimental results and effectiveness of the proposed method were addressed. Reviewer GPBC did not provide feedback on the authors’ rebuttal. The ACs feel that some of his/her concerns on the computational cost, the selection of hyperparameter d, and the use of symbol d may be addressed, but the concern on the poor performance on TextVQA may remain. Reviewer DasX did not provide feedback on the authors’ rebuttal either. The ACs feel that his/her concerns on how the failure modes of existing methods are solved and the insufficient analysis in the experiments can be addressed, while the concerns on the incremental contribution may be partly addressed.

**Reviewer Scores:**

From the author-reviewer discussions, the AC feels that reviewer d3yV may raise the score to 6 since most of his/her concerns were addressed, reviewer rmHq may keep his/her score as 4, reviewer GPBC may keep his/her score as 4, and reviewer DasX will keep his/her score as 6.
The overall score after rebuttal is likely to be 6, 4, 4, 6. Unfortunately, this paper cannot be accepted due to the competitiveness of ICLR.

---

### Decision · Program_Chairs · 2026-01-26

Reject